# On the Relation between Trainability and Dequantization of Variational Quantum Learning Models

**Elies Gil-Fuster**[*,1,2]    **Casper Gyurik**[3]    **Adrián Pérez-Salinas**[3,4]    **Vedran Dunjko**[3,5]

[1] Dahlem Center for Complex Quantum Systems, Freie Universität Berlin, 14195 Berlin, Germany

[2] Fraunhofer Heinrich Hertz Institute, 10587 Berlin, Germany

[3] $\langle aQa^L \rangle$ Applied Quantum Algorithms, Universiteit Leiden

[4] Lorentz Instituut, Universiteit Leiden, Niels Bohrweg 2, 2333 CA Leiden, Netherlands

[5] LIACS, Universiteit Leiden, Niels Bohrweg 1, 2333 CA Leiden, Netherlands

[*] emgilfuster@gmail.com

## Abstract

Quantum machine learning (QML) explores the potential advantages of quantum computers for machine learning tasks, with variational QML among the main current approaches. While quantum computers promise to solve problems that are classically intractable, it has been recently shown that a particular quantum algorithm which outperforms all pre-existing classical algorithms can be matched by a newly developed classical approach (often inspired by the quantum algorithm). We say such algorithms have been dequantized. For QML models to be effective, they must be trainable and non-dequantizable. The relationship between these properties is still not fully understood and recent works raised into question to what extent we could ever have QML models which are both trainable and non-dequantizable. This challenges the potential of QML altogether. In this work we answer open questions regarding when trainability and non-dequantization are compatible. We first formalize the key concepts and put them in the context of prior research. We introduce the role of "variationalness" of QML models using well-known quantum circuit architectures as leading examples. Our results provide recipes for variational QML models that are trainable and non-dequantizable. By ensuring that variational QML models are both trainable and non-dequantizable, we pave the way toward practical relevance.

## 1 Introduction

Machine Learning (ML) algorithms have been extensively researched and developed, resulting in numerous successful applications across various domains. Recently, there has been growing interest in Quantum Machine Learning (QML) (Schuld & Petruccione, 2021; Cerezo et al., 2021b; Bharti et al., 2022), which leverages quantum computing to potentially enhance traditional ML models. A significant focus in QML is on so-called variational QML models, where quantum operations are tuned by an optimizer, analogously to neural networks in classical ML. One of the main questions in the field at the moment is *how can we design good variational QML models?*

Addressing this question involves tackling several challenges, many of which mirror those encountered in the development of Neural Networks (NNs) for classical ML. Key considerations include properties derived from ML theory such as *expressivity* (Schuld et al., 2021; Pérez-Salinas et al., 2021), which refers to the model's ability to represent a wide range of functions; *generalization* (Caro et al., 2022; Gil-Fuster et al., 2024), the model's capability to perform well on unseen data; and *trainability* (McClean et al., 2018; Thanasilp et al., 2023), which concerns the ease with which a model can be optimized. Additionally, it is crucial to ensure both, that QML models are capable of solving *practically relevant* problems; and that their performance cannot be efficiently replicated by classical algorithms (Tang, 2019; Schreiber et al., 2023), a phenomenon dubbed as *dequantization*.

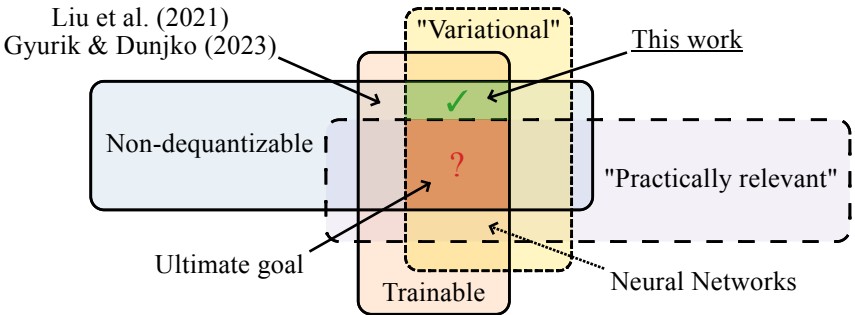

Figure 1: Depiction of research landscape. Dashed lines refer to concepts that are not well-defined. The dotted arrow reflects that we consider "variational" quantum machine learning models those which resemble neural networks in some sense. The inclusion of practical relevance remains an open topic.

Before we even start to think about discussing practical relevance, the questions surrounding dequantization and trainability must be resolved. Hence we focus on establishing precise language for three of the categories depicted in Fig. 1, namely: Non-dequantizable, trainable, and variational. The question of practical relevance may be the most interesting one, as it aligns with the search for sensible applications of quantum computing. However, it is also the most problematic characteristic to formally capture and treat theoretically. Our contributions pave the way towards eventually discussing practical relevance. We are guided by a strong wish to be as broad as possible while being sufficiently rigorous that mathematical proofs become possible.

Throughout this work we distinguish between QML models according to their operational resemblance to NNs. To this end we introduce a scale of "variationalness". Intuitively, what makes a model variational is a deep-layered structure (like NNs), and the trainable parameters appearing inside the model's layers. Variational QML models hold the promise of being broadly applicable, as well as being more efficient to train and to implement than quantum kernel methods (as is the case in classical ML).

Our first contribution is the formal definition of trainability and dequantization, to disentangle the discussion from related proxies. We next explain how trainability and non-dequantization are mutually compatible for non-variational QML models, based on prior works. We formally introduce a measure of variationalness and prove accompanying results to justify the need for a restricted notion of trainability. Finally, we resolve the open question regarding trainability and dequantization in variational QML: there do exist variational QML models which are gradient-based trainable and still non-dequantizable.

Our work provides clear outlook on the necessary future steps toward good variational QML models.

## 2 DEFINITIONS

In terms of notation, we call $\mathcal{X}$ the data domain, and $\mathcal{Y}$ the label co-domain. A learning task $(\mathcal{D}, R)$ is specified by a probability distribution $\mathcal{D}(\mathcal{X} \times \mathcal{Y})$ over inputs and labels, and a risk functional $R$. Given a function $f \colon \mathcal{X} \to \mathcal{Y}$, $R(f)$ measures the quality of $f$ as a solution to the learning task. We denote $\mathcal{D}_{\mathcal{X}}$ the marginal of $\mathcal{D}$ over the data $\mathcal{X}$. A learning model $(\mathcal{F}, \mathcal{A})$ consists of a family of functions $\mathcal{F}$ from $\mathcal{X}$ to $\mathcal{Y}$ and a learning algorithm $\mathcal{A}$. The learning algorithm takes as input a training set $S$ of labeled data and returns a function in the family $\mathcal{A}(S) \in \mathcal{F}$[1]. We use $N$ to denote the cardinality of $S$. We use $\hat{R}_S$ for the empirical risk, which is an estimator of $R$ restricted to the training set $S$. We ignore the possibility of a validation set for ease of presentation. We denote $f_{\mathcal{D}}^* \in \mathcal{F}$ an optimal solution to the task $(\mathcal{D}, R)$ within the model, and $f_S^* \in \mathcal{F}$ an optimum of the empirical risk $\hat{R}_S$ for the set $S$. If the function family is of the form $\mathcal{F} = \{f_\vartheta \colon \mathcal{X} \to \mathcal{Y} \mid \vartheta \in \Theta\}$, we talk about a parametrized learning model with parameters $\vartheta \in \Theta$. We use $\mathcal{P}(\Theta)$ for a probability distribution over the parameters.

---

[1]As a technicality, we define $\mathcal{F}$ as the range of $\mathcal{A}$ over all possible training sets, independently of their size.

## 2.1 PARAMETRIZED QUANTUM CIRCUITS FOR QML

We assume some familiarity with quantum computing, for more background in the formalism of quantum circuits, we refer readers to Appendix A. A Parametrized Quantum Circuit (PQC) is specified by a sequence of quantum gates depending on continuous parameters (Benedetti et al., 2019). The action of a PQC can be seen as a three step process: (1) *prepare* a fixed initial quantum state $\rho_0$, (2) transform the intial state into a different one $\rho_0 \mapsto \rho$ by applying a sequence of parametrized quantum gates, and (3) *measure* a fixed quantum observable $\mathcal{M}$. From PQCs we estimate the expectation value $\text{tr}(\rho\mathcal{M})$ (Nielsen & Chuang, 2000). Specifically for QML, we consider PQCs of polynomial *size* in the relevant scale of the learning task. The size of a PQC captures a combination of the number of qubits and the depth of the circuit. There are several approaches to designing PQCs, which make different use of the parameters of the circuit, but in every case the choice of underlying PQC is known to be critical.

A popular approach in variational QML is to consider a deep-layered PQC, where each layer may depend on both the input data $x \in \mathcal{X}$ and the trainable parameters $\vartheta \in \Theta$ (Pérez-Salinas et al., 2020). This way, the PQC results in a parametrized state $\rho(x; \vartheta)$. A real-valued labeling function arises from the expectation value of the fixed observable $\mathcal{M}$, as $f_\vartheta(x) = \text{tr}\{\rho(x; \vartheta)\mathcal{M}\}$. Each such PQC then gives rise to a hypothesis family $\mathcal{F}_Q = \{f_\vartheta(x) \,|\, \vartheta \in \Theta\}$. The resemblance between these PQCs and Neural Networks (NNs) comes from the layered appearance and the role of the trainable parameters.

Quantum kernel methods represent the main alternative to variational QML. Quantum kernel methods rely on quantum feature maps $x \mapsto \rho(x)$ (resulting in data-dependent parametrized states) to define a quantum kernel function $k(x, x') = \text{tr}\{\rho(x)\rho(x')\}$ (Schuld, 2021). Common quantum feature maps are realized as PQCs where the parameters encode the inputs $x$, and so quantum kernel functions also give rise to PQC-based QML models. For the quantum kernel approach, the resulting labeling functions are of the form $f_\alpha(x) = \sum_{i=1}^N \alpha_i k(x, x_i)$, where $\{(x_i)_i\}$ is the training set of size $N$ fed to the learning algorithm, and $\alpha = (\alpha_i)_i$ is a real vector of trainable parameters. Quantum kernel methods thus differ from other variational QML models in that their trainable parameters are not part of the PQC from where they stem, and often quantum feature maps do not display a layered-structure. Formally (see Appendix B for further details), the hypothesis family of such models is the set of real-valued linear maps of $\rho(x)$: $\mathcal{F}_\rho = \{f_\mathcal{M}(x) = \text{tr}(\rho(x)\mathcal{M}) \,|\, \mathcal{M} \in \text{Herm}\}$, Herm being the set of Hermitian operators of the appropriate dimension, exponential in the number of qubits. Notice we do not restrict the size of the training set when defining the hypothesis family of a learning model. Also, different Hermitian matrices can give rise to the same function if $\rho(x)$ is not full rank.

## 2.2 TRAINABILITY

Prior works have dealt with trainability in QML, but until now a formal definition has been lacking. Here we capture a few conflicting desiderata which we expose in detail in Appendix B.

**Definition 1** (Task-dependent trainability.). *Given a suite of learning tasks of interest $\{(\mathcal{D}_i, R_i)_i\}$, a learning model $(\mathcal{F}, \mathcal{A})$ is* trainable *if the following holds for each task: the empirical risk of the hypothesis produced by the learning algorithm is close to the optimal one within the class $\hat{R}_S(\mathcal{A}(S)) \approx \hat{R}_S(f_S^*)$ with high probability with respect to a training set $S$ of polynomial size being sampled i.i.d. according to $\mathcal{D}$.*

As a remark, notice our definition of trainability involves the empirical risk and not the expected risk. On the one hand, the empirical risk is the quantity we have access to. On the other hand, statements about the expected risk would arguably not be about trainability, but rather about generalization. In Section 3.1 we comment further on other similar notions.

We do not specify a notion of closeness for the risks, as this is an issue that is usually much easier to handle on a case-by-case basis. The conceptual idea is that a model is trainable if the labeling function outputted by the training algorithm is approximately optimal within the hypothesis family, for a set of tasks of interest. A (Q)ML model can still be considered trainable even if it displays poor performance in a task it is not well-geared to solve.

The discussion of useful notions of trainability does not end here, though. In particular, the definition allows for training algorithms $\mathcal{A}$ different from the ones QML practitioners would use. Accordingly, it makes sense to also discuss the nature of $\mathcal{A}$. For example, one could talk about *efficient trainability* by imposing that $\mathcal{A}$ must run in polynomial time in the size of the model and size of the training set. We could also distinguish between *quantum efficient* and *classical efficient* trainability.

## 2.3 BARREN PLATEAUS IN MACHINE LEARNING

The Barren Plateau (BP) phenomenon introduced in McClean et al. (2018) has been taken as a proxy for trainability in optimization problems. Here, we adapt the definition of BPs to make sense for any parametrized (Q)ML model, in a similar way to Thanasilp et al. (2023) and Barthe & Pérez-Salinas (2023). We propose a general definition which keeps the spirit that BPs are a concentration phenomenon over the variational parameters, independently of their roles. Given a parametrized (Q)ML hypothesis family $\mathcal{F}$ of size $n$, with variational parameters $\vartheta \in \Theta$, and a probability distribution $\mathcal{P}$ over $\Theta$, we define BPs in ML as follows:

**Definition 2** (Barren Plateaus in Machine Learning). *The hypothesis family $\mathcal{F} := \{f_\vartheta \,|\, \vartheta \in \Theta\}$ has a* Barren Plateau *with respect to the parameter distribution $\mathcal{P}$ and the data distribution $\mathcal{D}$ if the function values concentrate exponentially in the following sense:*

$$\mathbb{E}_{x \sim \mathcal{D}_{\mathcal{X}}}\left[\underset{\vartheta \sim \mathcal{P}}{\mathrm{Var}}\left[f_\vartheta(x)\right]\right] \in \mathcal{O}(\exp(-n)), \tag{1}$$

*where $\mathrm{Var}$ is the variance with respect to the parameter distribution $\mathcal{P}$.*

The data distribution $\mathcal{D}$ should be taken to be the one from the learning task we are tackling. In Appendix B, we discuss the operational meaning of BPs for different kinds of ML models.

## 2.4 DEQUANTIZATION AND SIMULATION

Dequantization can be understood in a number of ways, but here we adhere to a rather general notion in which given a quantum algorithm, there is a (perhaps related) classical algorithm with matching performance:

**Definition 3** (Dequantization in QML). *Given a suite of learning tasks of interest $\{(\mathcal{D}_i, R_i)_i\}$, a QML model $(\mathcal{F}_Q, \mathcal{A})$ is* dequantizable *if there exists a classical hypothesis family $\mathcal{F}_C$ and learning algorithm $\mathcal{A}_C$, such that the following holds for each of the learning tasks: given a training set $S \sim \mathcal{D}^N$, the performance of the classical learner is approximately at least as good as that of the quantum learner: $R(\mathcal{A}_C(S)) \lesssim R(\mathcal{A}(S))$.*

Although we define dequantization broadly in terms of a set of learning tasks, we wish to keep the focus on "dequantizing a model", and not on "dequantizing a problem", so the learning tasks of interest should be considered very general. Further discussion on this definition we defer to Appendix B.

One potential avenue for dequantization would be the ability to classically evaluate the functions $f_\vartheta \in \mathcal{F}_Q$ in the hypothesis family:

**Definition 4** (PQC simulation). *A PQC-based function family $\mathcal{F}_Q$ is* classically simulable *if there exists a classical poly-time algorithm which approximates $f_\vartheta(x)$ to good precision given $\vartheta$ and $x$.*

The definition can be further specialized to hold either for all inputs $x$, for all parameters $\vartheta$, or with high probability under respective distributions. Notice this is a statement only about a function family, and not about any learning task. Dequantization and simulation are essentially different properties in this way.

While we have focused on supervised learning tasks, recent milestone works have reported quantum advantage tasks involving random circuit sampling (Arute et al., 2019; Bluvstein et al., 2023). Further, works like Sweke et al. (2021) characterized the complexity of learning the distributions arising from random circuits, thus formalizing the problem as an unsupervised learning task. In this work we focus exclusively on supervised learning. An interesting future research direction is extending our formalism to also include the study of trainability and dequantization of unsupervised quantum learning models.

# 3 RELATED WORK

## 3.1 TRAINABILITY AND BARREN PLATEAUS

An important question remains open in previous works regarding the relation between trainability and (absence of) BPs. Works like (You & Wu, 2021; Anschuetz & Kiani, 2022) show that absence of BPs is not sufficient for trainability. Intuition in the field arguably dictates that the converse should not be true: presence of BPs should be sufficient for non-trainability. Later in Section 4, we show the opposite. We show how one can by-pass the problem of BPs with a learning algorithm that does not rely on local iterative optimization.

Our definition of trainability bears some similarity to notions of *agnostic PAC learning* (Haussler, 1992). The main difference to agnostic learning is that our definition involves the *empirical risk* $\hat{R}_S$, since in practice this is the quantity we have access to, and not the *expected risk $R$*. Further demanding that the output should be approximately optimal with respect to the expected risk would be a statement about generalization and inductive bias (Kübler et al., 2021), which is not our focus.

Another similarity is to the framework of *Meta-learning* (Schmidhuber, 1987), where a general learning model is trained to be able to later specialize and solve several different tasks. Although we defined trainability based on a set of tasks, we envision these to be taken to be very general while not containing all possible learning tasks, to avoid complexity theory pit-falls. In this sense, we consider different instances of the model being trained for each different task, and so our definition is essentially different to meta-learning.

## 3.2 DEQUANTIZATION AND SIMULATION

Our definition of dequantization captures general classes of quantum-inspired algorithms à la Tang (2019), as well as the recently introduced classical surrogate approaches (Schreiber et al., 2023; Rudolph et al., 2023). Our definition of simulation aligns with usual results (Bravyi & Gosset, 2016; Dias & Koenig, 2023).

In turn, Gyurik & Dunjko (2023) discuss many ways to establish quantum advantages in learning, which imply non-dequantization. Among the proposals we find the use of "non-classical" hypothesis families and also "non-classical" training algorithms, both of which can give rise to learning tasks that are hard for any classical learner and easy for a given quantum learner. They show that, in general, the simulation of a hypothesis family is neither sufficient nor necessary for dequantization of the corresponding QML model, as the training algorithm itself could perform classically-hard computations (see Section 3 in Gyurik & Dunjko (2023)). Conversely, it suffices to assume a classically-easy training algorithm for PQC simulation to be sufficient for dequantization of the QML model *for all learning tasks*. Works like Huang et al. (2021) and Schreiber et al. (2023) show that the converse is not true: there exist QML models which are dequantizable but whose circuits are not simulable.

Another relevant recent work is Cerezo et al. (2023), which discusses the relation between classical simulation and absence of BPs in optimization problems. There are two important differences between our setting and that of Cerezo et al. (2023). First, we distinguish between dequantization and simulation, and argue that the former is the relevant quantity. Second, we do not observe the possibility of quantum pre-computation in our definitions.

A novel research avenue asks whether specialized parameter initialization techniques known as *warm starts* (Puig-i-Valls et al., 2024) can break current attempts to classically simulate PQCs. In warm starts, one considers task-specific parameter distributions, or even data-dependent ones, which can result in non-simulable circuits which are actually simulable under the uniform distribution. The existence of other alternatives is currently being actively researched (Zhang et al., 2024).

## 3.3 TRAINABILITY AND DEQUANTIZATION OF EXISTING QML MODELS

In Section 2.1 we introduced two types of QML models: those based on deep-layered Parametrized Quantum Circuits (PQCs) and those based on kernel methods. Here we briefly comment on known results on trainability and dequantization for both types, and defer to Appendix C a deeper discussion of the two main examples we use in our results below. A central work in the quantum kernel literature is Liu et al. (2021), where a general-purpose quantum kernel was proposed that provably

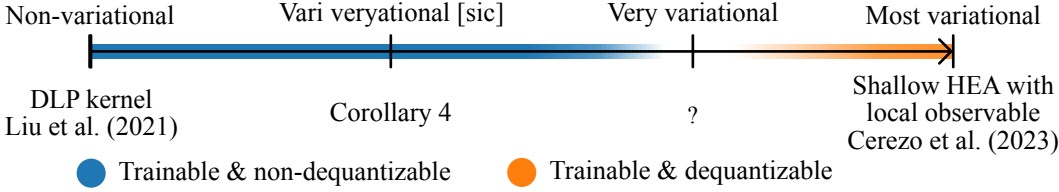

Figure 2: Spectrum of variationalness of quantum machine learning models considered in this work. The color indicates whether known trainable instances are also dequantizable across the spectrum. The shaded white area marks the possible existence of other trainable quantum machine learning models, strictly more variational than the vari veryational models we introduce, which are still non-dequantizable.

solves a task based on the Discrete Logarithm Problem (DLP), which is classically intractable under standard cryptographic assumptions. Under these assumptions, it follows that there exist kernel-based QML models which are trainable and non-dequantizable. Further, Thanasilp et al. (2022) characterized a phenomenon similar to BPs for kernel-based QML models called *vanishing similarity*. The main negative effect of this phenomenon is on the generalization of kernel-based QML models, and not on their trainability, for which no major issues have been identified so far. Finally, most variational QML models can be viewed as based on a family of PQCs called Hardware Efficient Ansatz (HEA) (Kandala et al., 2017) The HEA suffers from BPs (Kim et al., 2021) unless the PQC is shallow and the observable is local (Cerezo et al., 2021a). Intriguingly, Basheer et al. (2023) shows that the HEA is classically simulable as long as the PQC is shallow and the observable is local. From these works it follows that existing variational QML models based on the HEA are either trainable or non-dequantizable, but not both. In our main result below we show that this is not always the case: there exist variational QML models which are trainable and non-dequantizable.

## 4 TRAINABILITY VERSUS DEQUANTIZATION IN VARIATIONAL QML

In Section 3.3 we mentioned the DLP kernel as a proof that trainability does not imply dequantization for kernel-based QML models (Liu et al., 2021). In this section, we prove that the same holds for variational QML models, taking inspiration from the Hardware Efficient Ansatz (HEA) (Kandala et al., 2017). We first formalize a notion of *variationalness*, and next we show that there exist QML models which are variational in this sense, while also being trainable and non-dequantizable. We offer Fig. 2 as a schematic of the interplay between trainability, dequantization, and variationalness.

### 4.1 DEGREES OF VARIATIONALNESS IN QML

The starting point in the discussion is: what does actually make a quantum algorithm "variational"? The concept clearly has room for different interpretations, which affect what can be proven and what is true. The concept of variationalness is not well-defined, but all variational QML models can be seen as particular cases of HEAe, where some of the parametrized 2-qubit gates are either fixed or turned off. Accordingly we introduce a proxy notion that captures the "similarity to the HEA" as a measure of variationalness.

We build up to a list of well-posed properties that aim at capturing the essence of variationalness from the HEA design. We are interested in bottom-up, structural properties for the design of the PQC, and not abstract quantities that might be hard to estimate for a given circuit. We propose 5 properties that capture the essence of HEA. We refer to QML models that fulfill these properties as "vari veryational"[2]: (1) the PQC should consist of a single layer $U$ depending on both inputs $x$ and parameters $\vartheta$, that is applied sequentially several times $U(x; \vartheta_1), \ldots, U(x; \vartheta_L)$; (2) the number of layer applications $L$ should be tunable and independent of the number of qubits $n$; (3) the PQC should start from the $|0\rangle$ state; (4) we should measure a fixed observable $\mathcal{M}_0$, independent of the number of layers $L$, at the end; and (5) the parameters must be *gradient-based* trainable.

---

[2]"Vari veryational" is a *spoonerism* of "very variational".

We say a parametrized learning model is gradient-based trainable if it is trainable and the training algorithm is gradient based. We consider algorithms to be gradient based if they start from random parameters according to an initialization distribution $\mathcal{P}(\Theta)$ and then the algorithm is only allowed to iteratively update the parameters by exploring a small region around them at each step. This restricted notion of trainability has close ties to BPs. While it still does not hold that absence of BPs is sufficient for gradient-based trainability (You & Wu, 2021; Anschuetz & Kiani, 2022), now the converse does hold: absence of BPs is necessary for gradient-based trainability. Indeed, standard optimization theory dictates that, if a non-trivial optimization problem can be solved via gradient descent, then it follows that the gradient is not vanishingly small almost-everywhere (Goodfellow et al., 2016), and so it has no BPs in the sense of Definition 2. In Appendix E we offer a deeper insight into gradient-based trainability.

Vari veryational models contain deep-layered circuits, which are the main area of study of BPs. We do not call these models "very variational" in case there is a better definition. We invite researchers to contribute new bottom-up, structural properties to be added to our list of vari veryational models so that, ideally, we reach the best possible definition, which then would receive the name very variational.

We finish this section with two small results highlighting the need for a restricted notion of trainability (gradient-based trainability).

**Proposition 1** (Trainability with Barren Plateaus)**.** *There exist QML models which have BPs under the uniform distribution of parameters and are trainable for learning tasks that are classically intractable under standard cryptographic assumptions.*

*Proof sketch (full proof in Appendix D).* The main idea is to represent the functions arising from the DLP kernel as a HEA of polynomial depth, where only very specific choices of parameters recover the kernel. Then it holds that the HEA has a BP under the uniform distribution. At the same time, we can always fix the parameters of the HEA to recover the kernel, and this results in trainability from the trainability of kernels. The training algorithm here is not gradient based with respect to the parameters of the HEA, though, as first the optimal kernel parameters are found by an SVM and second the circuit parameters are fixed to recover the optimal kernel-based hypothesis. □

This small result confirms that *absence of BPs is neither necessary nor sufficient for trainability in general*. This does not mean that BPs do not suppose a problem when optimizing PQCs variationally, it only means they do not prevent all forms of training.

We also show that variational QML models can be trainable in general.

**Corollary 2.** *There exist deep-layered circuits, like Deep HEA, which are trainable and non-dequantizable.*

*Proof.* It follows directly from Prop. 1 and Liu et al. (2021). The training algorithm involves a reduction to kernel methods and is not gradient-based. The learning task is the same as the DLP kernel, which no classical algorithm can solve. □

## 4.2 Trainability and dequantization of vari veryational QML models

As our final contribution, we show that trainability does not imply dequantization for vari veryational models. We show this in two steps. We first propose a general recipe in Theorem 3 for gradient-based trainable, non-dequantizable QML models based on a computationally-hard function and an easy optimization task. We then adapt this recipe in Corollary 4 specifically for vari veryational QML models.

**Theorem 3** (Existence of trainable and non-dequantizable QML models.)**.** *Let $\mathcal{X} = \{0,1\}^n$ and $\mathcal{Y} = \{0,1\}$. Let $Q(x)$ be a function in* BQP *and not in* HeurBPP/poly *under a given distribution $\mathcal{D}_{\mathcal{X}}$, and let $U(x)$ be a unitary such that $\langle 0|U^{\dagger}(x)Z_1 U(x)|0\rangle = Q(x)$. Let $\mathcal{H}$ be a Hamiltonian, and $W(\vartheta)$ a parametrized unitary for which the following optimization problem can be solved with a given gradient-based algorithm $\mathcal{A}_W$:*

$$\vartheta^* \leftarrow \arg\max_{\vartheta \in \Theta} \langle 0|W^{\dagger}(\vartheta)HW(\vartheta)|0\rangle, \tag{2}$$

*and such that* $\max_{\vartheta \in \Theta} \langle 0|W^\dagger(\vartheta)HW(\vartheta)|0\rangle = 1$. *Call* $V(x;\vartheta) = U(x) \otimes W(\vartheta)$, *and* $\mathcal{M} = Z_1 \otimes H$, *and consider the corresponding hypothesis class* $\mathcal{F}_Q$:

$$\mathcal{F}_Q := \{f_\vartheta(x) = \langle 0|V^\dagger(x;\vartheta)\mathcal{M}V(x;\vartheta)|0\rangle \mid \vartheta \in \Theta\}. \tag{3}$$

*Let $\mathcal{D}$ specify a learning task:* $\mathcal{D}(x,y) = \mathcal{D}_\mathcal{X}(x)\delta(y = Q(x))$.

*Then $\mathcal{F}_Q$ is gradient-based trainable for $\mathcal{D}$, and it is not dequantizable.*

*Proof sketch (full proof in Appendix F).* Solving the learning task requires evaluating the function $Q(x)$. Since we take the function to be neither classically evaluatable nor classically learnable, it follows that no classical algorithm can solve the learning task. Since the function is in BQP and moreover we have access to the circuit that evaluates it, the quantum model can evaluate it. Then, the model is gradient-based trainable using the algorithm from the theorem statement. □

The construction in Theorem 3 resembles the construction in Appendix B of Cerezo et al. (2023), which also takes advantage of functions in BQP and not in HeurBPP/poly (under an appropriate distribution) to prevent classical simulation. The main difference is that we solve a supervised learning problem, instead of only evaluating the function. Note that the tensor product structure we use is clearly different from typical PQCs one may encounter in QML.

Neither the statement nor the proof of Theorem 3 mentions variationalness. It is in the following corollary that we construct a learning separation based on Theorem 3 but for which the quantum model is required to be vari veryational.

**Corollary 4** (Existence of trainable and non-dequantizable vari veryational QML model). *There exist vari veryational QML models which are gradient-based trainable and non-dequantizable with any number of layers up to sub-exponentially many in the number of qubits.*

*Proof sketch (full proof in Appendix F).* We must only take Theorem 3 and propose a special case where $\mathcal{F}_Q$ is vari veryational. The main hurdle is to ensure that $\mathcal{F}_Q$ allows a deep-layered structure, as in general even if a circuit $U(x)$ gives rise to a hard function, a sequential concatenation of the circuit $U^L(x) = U(x)U(x)\cdots U(x)$ could give rise to a non-hard function. A deep-layered structure is easy to achieve for the trainable part: first we can take $H$ to be single-qubit, with e.g. $H = Z$, and then we can take $W(\vartheta) = \prod_{j=1}^{L} R_X(\vartheta_j)$. With these, it follows that initializing $\vartheta$ uniformly at random and then performing gradient ascent is enough to solve the optimization task. For the classically-hard unitary, we start from any given $U(x)$ and expand it into a unitary $\tilde{U}(x)$ such that any number (up to exponential) of sequential applications of $\tilde{U}(x)$ is equivalent to a single application of $U(x)$. We combine $\tilde{U}(x)$ and $R_X(\vartheta_j)$ to reach a single layer $V^j(x;\vartheta_j) = \tilde{U}(x) \otimes R_X(\vartheta_j)$. It suffices for us to define $V(x;\vartheta) = \prod_{j=1}^{L} V^j(x;\vartheta_j)$ to fulfill all the assumptions of Theorem 3. By going through the list of properties of vari veryational QML models, we can see that indeed, this hypothesis family $\mathcal{F}_Q$ fulfills all of them. □

## 5 DISCUSSION

Throughout our results, we have made observations that may seen counter-intuitive at first glance. For example: absence of Barren Plateaus is neither necessary nor sufficient for trainability in general, but it is necessary for gradient-based trainability. Also: classical simulation is neither necessary nor sufficient for dequantization in general, but it is sufficient if we assume the training algorithm is classically efficient. These remarks call for a nuanced analysis when moving forward in future studies of trainability and dequantization in variational QML. In this work we discussed exclusively supervised learning scenarios, and we identify the outlook direction to expand our formalism to also include unsupervised learning tasks.

Already in Section 1 we identify practical relevance as our ultimate goal. Still, the models we have designed could be qualified as contrived, as for instance they involve PQCs split up into two disconnected parts. One may then worry that the proof of Theorem 3 provides limited insight to the design of PQCs for variational QML. This raises an open question: how can we take our constructions closer to solving practically relevant tasks? Following the statements in Cerezo et al. (2023), the

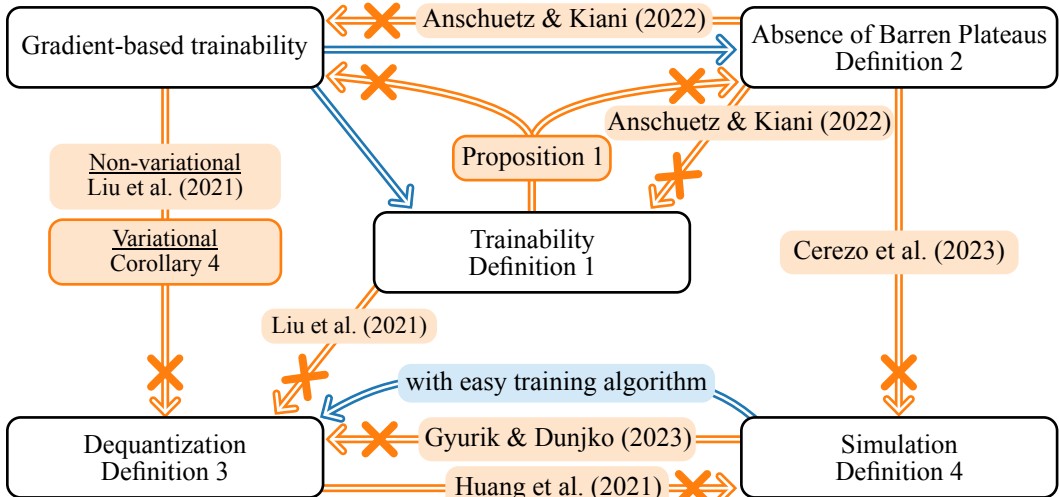

Figure 3: Summary of results. The blue lines are proven implications, the orange lines indicate the existence of counterexamples. Boxes with solid line refer to our contributions, boxes without solid line are results based on prior work.

road to practical relevance takes us away from most existing variational QML models, as all natural-looking ones to date are *either* trainable *or* non-dequantizable, but not both. This way, despite the contrived appearance, our constructions offer a basis from where to consider new approaches to the design of variational QML that can eventually mark all three boxes: trainability, non-dequantization, and practical relevance.

We next turn our attention to the broad applicability of our result. Theorem 3 is phrased in terms of an arbitrary function that is quantum-efficient and classically-intractable at the same time. Together with Corollary 4, our recipe can be made domain-specific for any single field where a central quantity is hard to learn and evaluate classically. It would be relatively straightforward to derive further Corollaries for example dealing with properties of quantum Hamiltonians that are known to be hard to infer classically. One could envision a situation where the effect of the variational parameters is to model a known source of noise, which while being classically tractable in isolation, cannot be characterized without a quantum computer when combined with the underlying quantity.

In a broader context, our results raise deeper questions. As discussed in Gyurik & Dunjko (2023), all learning separations we proved rely on computationally hard problems. Learning separations based on computational separations often result in statements that are arguably not about learning. In our case, the main bottleneck in the learning task was the ability to compute a classically-intractable function, but the learning itself was easy by construction. The way in which we ensured that our QML models would solve the problem was by planting the solution in the architecture of the hypothesis family itself. This may be deemed unsatisfactory, as the learning task could be trivially solved by the QML model we constructed. And yet, a critical question remains: how to design QML models with a certain inductive bias (Kübler et al., 2021; Peters & Schuld, 2022). It seems, then, that a fruitful research direction shall be to explore the space between completely generic circuits (no inductive bias) and circuits where the answer to a computationally hard problem is hard-coded.

On a different note, previous proofs of simulation of circuits that do not have BPs have relied to some degree on a specific class of simulation algorithms, exploiting so-called *polynomial subspaces* (Cerezo et al., 2023). An advantage of this approach is that the dequantization proofs come with a directly implementable algorithm. A disadvantage is that these simulation algorithms may only be guaranteed to succeed with high probability over the parameter domain, and that leaves the door open for them to fail precisely on the cases which would be interesting for QML. Our contributions in this direction show that it is easy to plant computationally-hard problems in variational QML, and so this challenges the role of average-case classical simulation algorithms.

## 6 CONCLUSION

In this work we have discussed general notions of trainability and dequantization for variational Quantum Machine Learning (QML) models. We have proposed clear definitions for trainability and dequantization. We have proved several relations between trainability, dequantization, and other common features in the literature like Barren Plateaus (BPs) and quantum circuit simulation, as depicted in Fig. 3. We have introduced a new family of *vari veryational* QML models, which convey the essence of commonly used deep-layered QML models and gradient-based training algorithms. We have resolved the open question whether, specifically for variational QML models, trainability and dequantization are mutually compatible. Our main contributions have been a clear formalization of the question, and a general recipe for variational QML models which are trainable and non-dequantizable.

The current main goal in variational QML is to find quantum advantage in learning using parametrized quantum circuits and practically relevant, real-life data, as sketched in Fig. 1. Our work arises in a specific context, where an alignment between trainability and dequantization for existing QML models has challenged the viability of the entire field (Cerezo et al., 2023). In this perspective, we resolve a critical open question: there do exist variational QML models which are trainable and non-dequantizable. We provide a recipe for QML models whose guiding principle is leveraging the classical hardness of evaluating some quantum functions. Specifically for the recent trend of studying the relation between BPs and classical simulability of quantum circuits, this offers a new conceptual perspective to the design of Parametrized Quantum Circuits (PQCs) for QML. The language we provide allows us to advance further, and enable formal discussion on the way toward practical relevance. From the point of view of QML practitioners, our prescription ensures that the resulting variational QML models are trainable and non-dequantizable. From here, variations on our recipe can be considered to eventually achieve good generalization and expressivity for specific domains of application.

### ACKNOWLEDGEMENTS

The authors thank Marco Cerezo, Jens Eisert, Zoë Holmes, Jarrod McClean, Luca Franceschi, and Ryan Sweke for their insightful comments in an earlier version of this draft. EGF thanks Greg White for pointing out the concept of spoonerism. EGF is a 2023 Google PhD Fellowship recipient and acknowledges support by the Einstein Foundation (Einstein Research Unit on Quantum Devices), BMBF (Hybrid), and BMWK (EniQmA). CG, APS and VD were supported by the Dutch National Growth Fund (NGF), as part of the Quantum Delta NL programme. This work was supported by the Dutch Research Council (NWO/OCW), as part of the Quantum Software Consortium programme (project number 024.003.03) and the project Divide & Quantum (with project number 1389.20.241) of the research programme NWA-ORC, and co-funded by the European Union (ERC CoG, BeMAI-Quantum, 101124342). Views and opinions expressed are however those of the author(s) only and do not necessarily reflect those of the European Union or the European Research Council. Neither the European Union nor the granting authority can be held responsible for them.

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

Supplementary Material for:
"On the Relation between Trainability and Dequantization of Variational Quantum Learning Models"

## A  BACKGROUND ON QUANTUM COMPUTING AND PARAMETRIZED QUANTUM CIRCUITS

Basic linear algebra lends itself nicely to formally discuss quantum computing. For further details, see Nielsen & Chuang (2000).

Quantum states can be thought of as generalizations of probability distributions over complex-valued fields. First, a discrete probability distribution can be represented as a diagonal matrix. Then, the set of all discrete probability distributions corresponds to the space of all diagonal matrices with positive entries and unit trace. In particular, these matrices are always *positive semi-definite* (PSD). Next, we can drop the requirement that matrices be diagonal, and we are left with quantum states $\rho$: Hermitian, PSD, unit-trace matrices,

$$\rho \in \text{Herm}, \rho \succeq 0, \text{tr}\{\rho\} = 1. \tag{A.1}$$

Hermitianity imposes real-valued diagonals, so the diagonal elements of a quantum state always form a discrete probability distribution. Further, the off-diagonal terms correspond to correlations that become relevant if one considers transformations of quantum states. The set of quantum states over $n$ qubits corresponds to matrices of dimensions $2^n \times 2^n$. Similarly, if we talked about classical bits, we would consider the space of probability distributions over bitstrings of length $n$, also $2^n$-dimensional. We reach quantum observables if we further drop the requirements of unit-trace and positive semi-definiteness. So, the set of all quantum observables over $n$ qubits is the set of all $2^n \times 2^n$ Hermitian matrices.

Quantum observables, as their name indicates, correspond to observable properties of the quantum states, as a direct generalization of observable properties of discrete probability distributions. Mathematically, the expectation value of a given quantum observable $\mathcal{M}$ with respect to a quantum state $\rho$ is captured by the Hilbert-Schmidt inner product of the corresponding matrices $\langle\mathcal{M}\rangle_\rho = \text{tr}\{\rho\mathcal{M}\}$ (there is no dagger due to the Hermitian property).

One way to conceptualize quantum computers is as machines that have a fixed initial quantum state $\rho_0$ and a fixed quantum observable $\mathcal{M}_0$, together with a set of possible operations $\mathcal{U}$ with which to furbish circuits. For instance, the trivial circuit would give rise to the value $\text{tr}\{\rho_0\mathcal{M}_0\}$. In a simple formulation, the allowed quantum operations map quantum states to quantum states as $\rho \mapsto U\rho U^\dagger$, for $U \in \mathcal{U}$, where $\mathcal{U}$ is a subset of the special unitary group $\mathcal{U} \subseteq \text{SU}(2^n)$. Then, a quantum circuit with $G$ gates $\{U_g\}_{g=1}^G$ where $U_g \in \mathcal{U}$, gives rise to the real value $\text{tr}\{U_G U_{G-1} \ldots U_1 \rho_0 U_1^\dagger \ldots U_G^\dagger \mathcal{M}_0\}$. We could call this a function from sequences of gates of length $G$ to the reals $\mathcal{U}^{\times G} \to \mathbb{R}$.

For *Parametrized* Quantum Circuits (PQCs), we include parametrized gates in the set of allowed operations. These are operations specified by a single number $\omega \mapsto U(\omega) \in \mathcal{U}$. A sequence of gates now involves both parametrized and non-parametrized gates. This is how we obtain a parametrized quantum state $\rho(\omega)$, where the actual transformations we apply to the initial state $\rho_0$ are ultimately specified by the values of the parameters $\omega$. A PQC gives rise to a function from real vectors to real numbers in that, after fixing the circuit layout, we obtain a different real value for each choice of parameters. This is where we have the chance of deciding the role of the parameters, to be either the input $x$, or the trainable ones $\vartheta$.

## B  MOTIVATION AND LIMITATIONS OF THE DEFINITIONS IN SECTION 2

### B.1  KERNEL-BASED HYPOTHESIS FAMILIES FOR ML

Given a quantum feature map $x \mapsto \rho(x)$, a corresponding quantum kernel $k(x, x') = \text{tr}\{\rho(x)\rho(x')\}$, and a training set $S = \{(x_i, y_i)\}_{i=1}^N$, usual kernel-based training algorithms output functions of the form $f_\alpha(x) = \sum_{i=1}^N \alpha_i k(x, x_i)$. Accordingly, the hypothesis family of such a model must be the

set of functions that can result from every possible training set $\bigcup_{S \in \mathbb{P}(\mathcal{X} \times \mathcal{Y})} \mathcal{A}(S) \subseteq \mathcal{F}_\rho$. In general, these are all functions of the form $\mathrm{tr}\{\rho(x)\mathcal{M}\}$, for any Hermitian matrix $\mathcal{M}$. Said otherwise, the set of functions realizable via quantum kernels is the set of linear functions of $\rho(x)$.

Usual kernel-based training algorithms make use of the representer theorem, which states that the optimal function in $\mathcal{F}_\rho$ to fit the training data lives in the span of training set $f_\alpha(x) = \sum_{i=1}^N \alpha_i k(x, x_i)$. This way, the training algorithm effectively constructs a subset of functions[3] within the hypothesis family, specified by the training set. This way, while the training algorithm sets up an optimization task involving $N$ parameters, the hypothesis family itself remains of dimension exponential in the number of qubits.

## B.2 TRAINABILITY

Our guiding principle is to propose a definition of trainability that captures practically relevant scenarios. For example, we should conclude that deep neural networks are often trainable. At the same time, a precise definition cannot ignore the fact that finding global optima of even simple learning models is NP-hard and too much to ask for (Pfister & Bresler, 2018), as we elaborate shortly. Recall that we refer to "a learning model" $(\mathcal{F}, \mathcal{A})$ as a pair formed by a hypothesis family $\mathcal{F}$ and training algorithm $\mathcal{A}$. We discuss notions of trainability that involve both at the same time. Consequently, we want to capture the notion that a learning model is considered trainable if the solutions within $\mathcal{F}$ found by $\mathcal{A}$ are *good enough*. Two aspects need be pinned down: what does good enough mean, and which tasks should the solutions be good for.

Venturing a naive definition like "a model is trainable if the learning algorithm solves every given task perfectly" runs into fundamental issues: under this definition no model can ever be trainable due to the no-free-lunch theorem (Wolpert & Macready, 1997), which guarantees that for any model there is at least one learning task in which it must fail. We could next lower the requirement from "the learning algorithm solves every given task perfectly" to "the learning algorithm solves every given task approximately", meaning "the performance of the hypothesis reached by the learning algorithm $R(\mathcal{A}(S))$ is approximately optimal *within the hypothesis class* $R(\mathcal{A}(S)) \approx R(f_\mathcal{D}^*)$, for any task $\mathcal{D}$". Under this definition we know that neither classical neural networks nor PQCs already of logarithmic depth can be trainable, unless $\mathsf{P} = \mathsf{NP}$ (Bittel & Kliesch, 2021). Hence we must relax the assumptions further. Responding with a more relaxed definition like "a model is trainable if there exists a learning task for which the learning algorithm outputs an approximately optimal hypothesis" results in too weak a notion for which all models are trainable, as another consequence of the no-free-lunch theorem. Basing the definition on reaching locally optimal solutions is no help either, as there are models for which almost all local optima are guaranteed to be as bad as random guessing (Anschuetz & Kiani, 2022), so there would be models which are trainable under this definition, but which always output bad solutions. Many known QML models display mostly bad local minima, and it is a still poorly understood miracle of deep learning that this it so often *not* the case there, where empirically good local minima seem to be easy to find. To avoid these problems, we decide to work with a context-dependent notion of trainability.

**Definition 1** (Task-dependent trainability). *Given a suite of learning tasks of interest $\{(\mathcal{D}_i, R_i)_i\}$, a learning model $(\mathcal{F}, \mathcal{A})$ is* trainable *if the following holds for each task: the empirical risk of the hypothesis produced by the learning algorithm is close to the optimal one within the class $\hat{R}_S(\mathcal{A}(S)) \approx \hat{R}_S(f_S^*)$ with high probability with respect to a training set $S$ of polynomial size being sampled i.i.d. according to $\mathcal{D}$.*

The main limitation of this definition is that it can be difficult to apply in real-life scenarios. Indeed, knowing the performance of the empirical risk minimizer is a necessary condition for assessing the trainability of learning model under this definition, which can be difficult. Also, for broad applicability of the definition, one would like to decide whether a model is trainable without having to train it, in order to efficiently decide between different models to use.

---

[3]Being careful, we must *not* call this a hypothesis family, because of its training-set dependence.

### B.3 BARREN PLATEAUS

The *Barren Plateau* (BP) (McClean et al., 2018) phenomenon appeared in the literature of Variational Quantum Algorithms (VQAs) (Cerezo et al., 2021b; Bharti et al., 2022). The main idea for VQAs is to tune the control parameters of a parametrized quantum circuit with a hybrid quantum-classical optimization loop. Such an approach requires computing gradients of the function to be optimized. The boon of this approach is that one has access to the large Hilbert space of quantum computing via self-adjusting classical control knobs. The bane of this approach is that for large systems, the gradients become smaller than the available quantum machine precision. Plenty of literature has been devoted to charting the space of VQAs as to whether they suffer from BPs, ways to avoid BPs, and sharpening the analytical tools required to diagnose BPs (Larocca et al., 2024).

One can think of QML as a special case of a VQA, where the function to be optimized happens to also depend on training data. Still, the common definition of BPs anticipates a scenario where there are only optimization parameters. We would like to introduce a definition of BPs that is applicable to any *parametrized* learning model[4], be it quantum or else. For this reason, we introduce a notion of BPs that relies on the concentration of measure: functions coming from the same parametrized hypothesis family should concentrate to the same function even if they are realized from different parameter specifications.

**Definition 2** (Barren Plateaus in Machine Learning). *The parametrized hypothesis family* $\mathcal{F} :=$ $\{f_\vartheta \mid \vartheta \in \Theta\}$ *has a* Barren Plateau *with respect to the parameter distribution* $\mathcal{P}$ *and the data distribution* $\mathcal{D}$ *if the function values concentrate exponentially in the following sense:*

$$\mathop{\mathbb{E}}_{x \sim \mathcal{D}_\mathcal{X}} \left[ \mathop{\mathrm{Var}}_{\vartheta \sim \mathcal{P}} [f_\vartheta(x)] \right] \in \mathcal{O}(\exp(-n)), \tag{B.1}$$

*where* $\mathrm{Var}$ *is the variance with respect to the parameter distribution* $\mathcal{P}$, *and* $\mathcal{D}_\mathcal{X}$ *is the marginal distribution over* $\mathcal{X}$ *arising from the problem distribution* $\mathcal{D}$.

Indeed, this definition fulfills the two target properties: it captures a concentration phenomenon, and it can be applied to any parametrized learning model. The operational meaning of this definition changes depending on the type of the ML model, though. We set this definition so that the moral of the story remains the same: if a ML model exhibits a BP in this sense, then there may be obstacles preventing trainability via usual methods.

Discussing whether kernel-based QML models have BPs highlights their fundamental differences from other variational QML models. As introduced in Section 2, the hypothesis family $\mathcal{F}_\rho$ of a model based on the quantum kernel arising from $\rho(x)$ is the set of real-valued linear functions of $\rho(x)$. Then, for us to ask whether $\mathcal{F}_\rho$ has a BP, we must provide a distribution over the space of Hermitian matrices. This requirement already diverges entirely from the typical pipeline of kernel methods for (Q)ML.

For instance, the uniform distribution over all Hermitian matrices of bounded norm would result in exponentially small function values independently of $\rho(x)$. That does not mean that the output functions of kernel-based training algorithms are exponentially concentrated in expectation. Rather, this highlights that Barren Plateaus do not affect all possible forms of training.

The work of Thanasilp et al. (2023) performs a similar analysis. By introducing the concept of *vanishing similarity* (a concentration of the kernel function over random inputs), they offer a platform to study the learning dynamics of gradient-based training algorithms. Here we concentrate on the hypothesis family, and not on the data-dependent function family produced by the training set. We note that there could be different natural definitions of BPs for QML, and that specifically for kernel-based hypothesis families the presence of BPs would be very sensitive to small changes in the definitions.

Notice that under our definition, a hypothesis family with only constant functions would not have a BP as long as the value of the constant has a high variance with respect to the model parameters.

---

[4]Recall that we defined parametrized learning models as those whose functions are specified by at most polynomially-many real values.

### B.4 DEQUANTIZATION AND SIMULATION

We give a definition of dequantization that is task-dependent, and is set aside from common definitions of circuit simulation. The guiding principle was to lay out the scenario in which one would not need to use a quantum computer in order to solve a learning task.

**Definition 3** (Dequantization in QML). *Given a suite of learning tasks of interest $\{(\mathcal{D}_i, R_i)_i\}$, a QML model $(\mathcal{F}_Q, \mathcal{A})$ is* dequantizable *if there exists a classical hypothesis family $\mathcal{F}_C$ and learning algorithm $\mathcal{A}_C$, such that the following holds for each of the learning tasks: given a training set $S \sim \mathcal{D}^N$, the performance of the classical learner is approximately at least as good as that of the quantum learner: $R(\mathcal{A}_C(S)) \lesssim R(\mathcal{A}(S))$.*

The alternative would be the scenario in which one does not need to use a quantum computer to evaluate expectation values.

**Definition 4** (PQC simulation). *A PQC-based function family $\mathcal{F}_Q$ is* classically simulable *if there exists a classical poly-time algorithm which approximates $f_\vartheta(x)$ to good precision given $\vartheta$ and $x$.*

As justification, we saw that there is a gap between these definitions in Section 3.

A limitation of this definition is that it could allow for very powerful learning models to be dequantized for relatively easy learning tasks. Indeed, if we started from learning tasks which a classical model can solve approximately optimally to begin with, then every all-powerful learning model is immediately dequantizable for those learning tasks, even if the classical model is completely unrelated to the structure of the all-powerful model. This is in contraposition for instance to the dequantization techniques proposed by Tang (2019), where a clever classical data structure was able to dequantize the proposed QML approaches by replicating their information processing pipeline. Conversely, we would say "the DLP kernel has been dequantized for the DLP-based learning task" already if we found an efficient classical algorithm for the discrete logarithm, in which case learning would be done by a completely different classical method, and not necessarily a kernel-based one. Thus this definition spans also a gray area housing counter-intuitive true statements. This is not to our detriment, as it is our manifest intent to consider broad definitions that could be further specialized a posteriori if needed.

## C FURTHER DISCUSSION ON TRAINABILITY AND DEQUANTIZATION OF EXISTING QML MODELS

In this section we first provide a deeper discussion on the two models that appear in our results: a quantum kernel linked to the Discrete Logarithm Problem (DLP) Liu et al. (2021), and the Hardware Efficient Ansatz (HEA) Kandala et al. (2017). At the end we comment on how other existing QML models fit our categories of trainability and dequantization.

### C.1 THE DLP KERNEL

Liu et al. (2021) used the classical hardness of the Discrete Logarithm Problem (DLP) to show a quantum-classical separation in learning. They designed a learning task that cannot be solved by any classical learner according to standard cryptographic assumptions. They showed that the same task could be solved using a general-purpose quantum kernel, which we call the "DLP kernel" from now on. We next analyze the DLP kernel in terms of its trainability and dequantization. For further details on the DLP kernel we refer readers to Liu et al. (2021). Additionally, Appendices B and E contain thorough discussions on BPs and gradient-based trainability specifically for kernel methods.

First, the DLP kernel is trainable because the representer theorem guarantees that all kernel methods are trainable (Schölkopf & Smola, 2002). For instance, using a Support Vector Machine (SVM) we obtain the actual minimum of the empirical risk. Also, Thanasilp et al. (2022) identified that kernel-based ML models could suffer from a phenomenon similar to BPs, dubbed *vanishing similarity*, if the kernel function $k(x, x')$ is exponentially close to 0 almost everywhere. In Liu et al. (2021) we see that the DLP kernel takes non-trivial values, so the DLP kernel does not suffer from vanishing similarity.

In addition, the learning-theoretic results from Liu et al. (2021) guarantee that no classical algorithm can solve the same learning task as the DLP kernel efficiently. This proves that the QML model based on the DLP kernel is non-dequantizable. The latter argument says nothing about simulating the corresponding circuit directly, but since *simulable* $\Rightarrow$ *dequantizable* with a classically-easy training algorithm (like SVM), it immediately follows that *non-dequantizable* $\Rightarrow$ *non-simulable*.

It thus follows from prior work that the DLP kernel is both trainable and non-dequantizable.

## C.2 VARIATIONAL QML WITH THE HEA

The Hardware Efficient Ansatz (HEA) (Kandala et al., 2017) is a circuit template for universal quantum computation. The HEA is composed of sequential layers that respect the lower-level connectivity of the computing platform. A particularly well-studied version is the 1-dimensional HEA, where qubits are organized in a line and each qubit is only connected to its nearest neighbors. Such 1-dimensional HEAe[5] give rise to deep-layered circuits, where each layer is a column of 2-qubit gates, and consecutive layers alternate between even and odd connections. HEA is a natural substrate for variational QML models as introduced in Section 2: we first prepare a parametrized state $\rho(x; \vartheta)$ and next measure a fixed observable $\mathcal{M}$. For further details on the HEA we refer readers to Kandala et al. (2017), and specifically to Cerezo et al. (2021a) for a discussion on its Barren Plateau analysis. Importantly, the free parameters of $\vartheta$ are typically trained *variationally*, via an iterative local optimization procedure. We next characterize the trainability and potential for dequantization of HEA-based QML models.

Cerezo et al. (2021a) showed that HEAe can have BPs under the uniform distribution of parameters due to their generic structure. In parallel, empirically HEAe have defied being trained (Kim et al., 2021). A special case is that of HEAe based on shallow PQCs, of up to logarithmic depth. Cerezo et al. (2021a) proved that shallow HEAe do not have a BP when the observable is local (it acts only on a few qubits). Similarly Basheer et al. (2023) showed that HEAe are classically simulable provided the circuits are shallow and the observable is local. Thus, together with a classically-tractable training algorithm, shallow HEAe with local observable are dequantizable. Crucially, while generic HEAe of linear depth or shallow HEAe with global observables are not classically simulable, training them by usual means is expected to be difficult due to BPs.

Indeed, the same conditions that ensure absence of BPs in HEAe also allow for their dequantization. Intriguingly, this is not an isolated case: Cerezo et al. (2023) shows that for most PQC Ansätze for which we have a proof of absence of BPs, we also have a proof of classical simulation. This hints at a larger question: *Does absence of BPs imply classical simulation for all variational quantum circuits?* The answer is negative in general, as one can construct contrived counter-examples (see Appendix B of Cerezo et al. (2023)). Still, the question of absence of BPs versus classical simulation in naturally-occurring quantum optimization problems is not fully resolved. Comprehensive sufficient and necessary conditions for absence of BPs to imply dequantization are not yet known.

Also, the question of trainability versus dequantization of variational QML models has not been previously addressed.

## C.3 OTHER EXISTING QML MODELS

We again separate two types of QML models: those based on kernel methods and those based on PQCs with local optimization methods.

For kernel-based models, the discussion in Thanasilp et al. (2022) covers most of the ground. They explain that trainability is not a central issue in kernel-based ML, both quantum and not, as the training algorithms used in practice are empirical risk minimizers. We defined trainability as the ability of the training algorithm to output a close-to-optimal solution with respect to the empirical risk. It is generally true that kernel-based (Q)ML models fulfill this condition, so they are generically trainable. Regarding dequantizability, not many results have proven quantum advantage in learning using a kernel-based model beyond Liu et al. (2021) and Huang et al. (2021). An alarm raised in Thanasilp et al. (2022) is that, for QML models based on generic quantum kernels, one expects an alignment between "lack of vanishing similarity" and classical simulation of the PQC. In this case

---

[5]The plural of Ansatz is Ansätze.

lack of vanishing similarity is linked to good generalization performance, so the trade-off becomes between generalization and non-dequantization, which is also negative. It remains an open question whether the general-purpose DLP kernel is the only one which formally fulfills both trainability and non-dequantization.

Most existing variational QML models can be seen as special cases of the HEA, where different models arise from different choices of which gates are fixed to be data-dependent, and which gates are left open for their parameters to be optimized. In this sense, one would expect generic variational QML models to suffer from BPs whenever the HEA they are based on suffers from a BP. This effect could be mitigated either by considering very specific circuit structures, or by turning several gates completely off, or by introducing correlations between the parameters that would break the i.i.d. assumptions in the theorems that diagnose BPs. This way, regardless whether encoding first (Farhi & Neven, 2018), data re-uploading Pérez-Salinas et al. (2020), or flipped models Jerbi et al. (2023), the trainability of variational QML models could be thwarted by the propensity of HEA to suffer from BPs. The picture is similar for dequantization: if the underlying PQC is classically simulable, then generically the resulting QML is also simulable, relatively independent of the type of variational QML model at hand. A potential cure against BPs is the fact that the data distribution is unknown. Parametrized function families have BPs according to a specific distribution. In supervised learning, we typically assume the data distribution is fixed but unknown. So, formally this means that we cannot establish analytically whether a variational QML model suffers from a BP unless we impose certain assumptions on the data distribution. Noteworthy is also that recent works have specialized the study of BPs to data re-uploading circuits (Barthe & Pérez-Salinas, 2023; Mhiri et al., 2024), via the Fourier picture introduced in Schuld et al. (2021).

## D PROOF OF PROPOSITION 1

Here we restate and prove the proposition in Section 4, using the definitions for trainability and Barren Plateaus (BPs) introduced in Section 2.

**Proposition 1** (Trainability with Barren Plateaus). *There exist QML models which have BPs under the uniform distribution of parameters and are trainable for learning tasks that are classically intractable under standard cryptographic assumptions.*

*Proof.* We prove the statement directly by giving an example. Consider the DLP kernel $k_{\mathrm{DLP}}(x, x') = \mathrm{tr}(\rho(x)\rho(x'))$ introduced in Liu et al. (2021). Given a training set $\{(x_i, y_i)\}_{i=1}^N$, consider the set of functions resulting from using the representer theorem:

$$\mathcal{F}_k := \left\{ f_\alpha(x) = \sum_{i=1}^N \alpha_i \, \mathrm{tr}\{\rho(x)\rho(x_i)\} \,\middle|\, \alpha \in \mathbb{R}^N \right\}. \tag{D.1}$$

To evaluate $f_\alpha(x)$, we typically evaluate each overlap $\mathrm{tr}\{\rho(x)\rho(x_i)\}$ separately and then sum. We could instead evaluate it all at once with a single circuit, using the Linear Combination of Unitaries (LCU) formalism (Childs & Wiebe, 2012). Here, we use LCU to design a circuit whose observable corresponds to a mixed quantum state, which we express as a classical mixture of pure states explicitly. In this way, we do not take advantage of the interference phenomena that LCU usually allows, but rather use it to create a linear combination of states.

We consider the parametrized observable $\mathcal{M}(\alpha, (x_i)_i) = \sum_{i=1}^N \alpha_i \rho(x_i)$ and we recognize we can implement it as an LCU circuit. On the one hand, we prepare the state $|\alpha\rangle = \sum_i \alpha_i |i\rangle$ on an auxiliary system. W.l.o.g. the vector $\alpha$ can be taken to be appropriately normalized. On the other hand, we consider the data-dependent unitary gate $V(x)$ that gives rise to the quantum embedding $\rho(x) = V(x)|0\rangle\langle 0|V^\dagger(x)$. Then, we construct a circuit in which, after $|\alpha\rangle$ has been prepared, we apply $V(x_i)$ on the work register, controlled on the auxiliary register being in state $|i\rangle$. We initialize the computer to be on the all $|0\rangle$ state, then evolve it to become $|\alpha\rangle \otimes V(x)|0\rangle$, then apply the controlled-$V(x_i)$ gates, and finally measure the projector onto $|0\rangle$ on the work register and a diagonal observable that takes care of the signs on the auxiliary register. The resulting expectation value is the same as for the kernel method $\mathrm{tr}\{\rho(x)\mathcal{M}(\alpha, (x_i)_i)\} = f_\alpha(x)$.

In the LCU circuit, some gates are 1-qubit trainable, and some gates are 2-qubit fixed, but we have an efficient description of the whole circuit. Next, rewrite all gates as 2-qubit arbitrary gates,

with a given known parametrization, from which we can recover the LCU circuit explicitly. The 2-qubit gates at this step must observe a 1-dimensional connectivity graph, so each qubit may only be connected to adjacent qubits. This means that any non-local 2-qubit gate must be compiled as linearly many local 2-qubit gates. Organize all gates in non-commuting layers, and complete any layers with new 2-local gates, overall resulting in a deep-layered brickwork architecture. This step includes also the encoding gates required to prepare $\rho(x)$, where a possibly-discrete gate set is compiled with the continuous set of arbitrary local 2-qubit gates.

We call $\vartheta$ the parameters of the brickwork Ansatz, for which, crucially, we have an efficient specification $(\alpha; (x_i)_i) \mapsto \vartheta$ such that we recover the LCU kernel-based parametrized observable. We choose not to write $\vartheta(\alpha; (x_i)_i)$ for ease of notation. The depth of the circuit is at least linear in the size of the training set, and also at least linear in the required complexity of preparing $\rho(x)$. The observable we measure at the end is global: it consists of a collection of Pauli-$Z$s and projectors onto the $|0\rangle$ state on all qubits. Then, from Cerezo et al. (2021a) we know that this quantum circuit has a BP under the uniform distribution over $\vartheta$, as it is an instance of the HEA with linear depth and global observable. Crucially, this Ansatz has a BP with respect to the circuit parameters $\vartheta$, and not the free parameters of the kernel-based functions $\alpha$, which are just a subset of the entire function family, and have a very particular parametrization associated to it.

Nevertheless, from Liu et al. (2021) we know this model is capable of solving a learning task based on the DLP which is assumed to be hard for any classical learner. The reduction from this model onto that of Liu et al. (2021) needs only that we fix the parameters to recover the LCU circuit. So, the learning algorithm we use takes two steps:

1. Use the $k_{\mathrm{DLP}}$ and a Support Vector Machine (SVM) to reach the optimal $\alpha^*$ vector, which we know actually solves the problem as explained in Liu et al. (2021).

2. Set the parameters $\vartheta$ of the brickwork Ansatz to recover $f_{\alpha^*}(x)$ as an LCU circuit.

The solution we are left with $f_{\alpha^*}$ satisfies the requirements of the proposition: it is trainable, in the sense that there is a training algorithm with which it solves a task of interest; and it has a BP, in the sense that the cost function concentrates exponentially with respect to the uniform distribution of the circuit parameters. Crucially, we say the model is trainable because, even though the hypothesis family generated by the brick-layered Ansatz is strictly larger than that generated only by the quantum kernel, the representer theorem guarantees that the solution found by SVM is optimal over all linear models, of which anything generated by the layered Ansatz is a special case. This completes the proof.

We note the same strategy would not necessarily follow from any arbitrary trainable circuit: in the sense that just compiling a trainable PQC into a deep-layered brickwork Ansatz because the added functions might give rise to a better solution to the learning task. It could be that by making the hypothesis family larger, what used to be a good-enough solution for the smaller model stops being close to optimal for the larger model. For this step to still work, we must exploit some guarantee of optimality like the one given by the representer theorem. $\qquad\square$

The goal of this result is to reinforce the difference between the notions of trainability that we introduce in this work and its proxy, Barren Plateaus. Prop. 1 only points out that trainability and BPs are phenomena of different nature.

## E   GRADIENT-BASED TRAINABILITY

Trainability as introduced in Definition 1 is very broad, since it allows for unconventional training algorithms unused in practice. We introduce *gradient-based* training algorithms with the goal of reaching a more stringent notion that relates to currently used training algorithms (Cerezo et al., 2021b; Bharti et al., 2022; Nietner, 2023). In the language of Section 2, for a model $(\mathcal{F}, \mathcal{A})$ to be gradient-based trainable, it needs to be trainable according to Def. 1, and the training algorithm $\mathcal{A}$ needs to be a gradient-based. This section introduces and motivates gradient-based training algorithms and discusses the relation between gradient-based trainability and BPs.

We draw inspiration from the well-known Gradient Descent (GD) algorithm, which requires an initial specification of parameters, and uses the gradient of the empirical risk with respect to the parameters $\nabla_\vartheta \hat{R}_S(f_\vartheta)$. In GD an initial parameter specification is typically sampled from a distribution $\mathcal{P}$ over the parameter domain $\Theta$. Then, parameters are sequentially updated in the direction opposite to the gradient $\nabla_\vartheta \hat{R}_S(f_\vartheta)$ to minimize the empirical risk. The particular parametrization $\vartheta \mapsto f_\vartheta$ is relevant, as different parametrizations of the same hypothesis family give rise to different gradients, and thus to different outputs using GD (Wiersema et al., 2024). This way, whether GD produces a good output from a hypothesis family $\mathcal{F}$ depends not only on the task $(\mathcal{D}, R)$, but also on the initialization distribution $\mathcal{P}$, and the specific parametrization.

To paint a concrete picture, we offer an algorithmic description of a general gradient-based training algorithm in Algorithm 1. The following example retains the central role of both: the initialization distribution $\mathcal{P}$ and the specific parametrization of $\mathcal{F}$[6].

---

**Algorithm 1** Gradient-based training algorithm.

---

**Input:** $\mathcal{F} \coloneqq \{f_\vartheta \mid \vartheta \in \Theta\}$       ▷ Parametrized hypothesis family.
**Input:** $\hat{R}_S(f_\vartheta)$       ▷ Empirical risk functional.
**Input:** $\mathcal{P}(\Theta)$       ▷ Parameter initialization distribution.
**Input:** $C(\vartheta) \leftarrow (\nabla_\vartheta \hat{R}_S(f_\vartheta), H_\vartheta \hat{R}_S(f_\vartheta))$       ▷ Learning rate.
**Output:** $\vartheta \in \Theta$       ▷ Trained parameters.
 1: $\vartheta_0 \sim \mathcal{P}$       ▷ Sample initial parameters
 2: **for** $t$ in $\{1, \ldots, T\}$ **do**:
 3:     $\vartheta_t \leftarrow \vartheta_{t-1} + C(\vartheta_{t-1})\nabla_\vartheta \hat{R}_S(f_{\vartheta_{t-1}})$       ▷ Update rule.
 4: **end for**
 5: **return** $\vartheta_T$

---

An important restriction of gradient-based training algorithms is that they should not be allowed to evaluate the empirical risk corresponding to arbitrary parameters $\vartheta$. Rather, the algorithm should start from a random initial specification $\vartheta_0 \sim \mathcal{P}$, and then only be allowed to iteratively select new parameters which are geometrically close in parameter space. This is why we limit the algorithm to only take steps in the direction of the gradient, up to an adaptive learning rate $C(\vartheta)$. We prevent more complex algorithms to hide inside this update rule by restricting the allowed functional form of $C(\vartheta)$, which at step $t$ may only depend on $t$, $\nabla_\vartheta \hat{R}_S(f_{\vartheta_t})$, and $H_\vartheta(\hat{R}_S(f_{\vartheta_t}))$, where $H_\vartheta$ denotes the Hessian.

We note that a parameter distribution $\mathcal{P}$ already played a central role when introducing Barren Plateaus (BPs) in Def. 2. Gradient-based trainability and BPs are clearly deeply related: if the model has a BP according to the distribution $\mathcal{P}$, then the value of all functions concentrates exponentially with high probability over $\mathcal{P}$, and so the gradient becomes exponentially small with high probability over $\mathcal{P}$. Having exponentially small gradients signifies a much harder problem for QML than for classical ML models. For QML models, the available precision is only polynomial in the total quantum runtime (or said otherwise, exponential precision requires exponential quantum time). Conversely, for classical ML models, precision is exponential in the number of bits of memory, and the total runtime is polynomial in the number of bits, so exponential precision is within budget. All together: having exponentially small gradients already represents a problem by itself, as the time to convergence could be exponentially long. But furthermore, for QML models we would not be able to resolve the exponentially small gradients in polynomial time, so exponentially small gradients are statistically indistinguishable from 0 gradients.

We next briefly comment on the gradient-based trainability of learning models based on quantum kernels. The defining factor of these learning models is that the training algorithm uses kernel-based techniques. Kernel-based training algorithms invoke the representer theorem to build an effective, data-dependent function family over which to optimize a few parameters. Since this step is not in-

---

[6]There exist non-gradient-based methods which share these features, like genetic or evolutionary algorithms, which we could have also included in an even more general family. We decide against a more general notion for the ease of presentation.

cluded among the allowed operations of gradient-based training algorithms, it follows that in general these models are not gradient-based trainable.

Said differently, kernel-based training algorithms first restrict the search space to a subspace given by data, and then optimize within the subspace. Often, this second optimization can follow the gradient-based prescription, which would be efficient. However, the first step (restriction to a data-dependent subspace) is something other than "random parameter initialization plus local optimization". In all, even though kernel-based training algorithms could have a gradient-based optimization subroutine, we demand for an algorithm to be end-to-end gradient based in order to qualify for our definition.

As we introduced, the hypothesis family of learning models based on quantum kernels is the set of linear models of the feature map $\text{tr}\{\rho(x)\mathcal{M}\}$, for $\mathcal{M} \in \text{Herm}$. On the one hand, usual risk functionals could be convex in the space of linear models, which would in principle open the door to gradient-based trainability. On the other hand, the expected runtime of such an algorithm would be at least linear in the dimension of the space. Given that $\text{Herm}$ is of dimension exponential in the number of qubits, these models are in general not *efficiently* gradient-based trainable.

In general, it may well be that a given parametrized learning model has a BP or is gradient-based trainable under a given distribution, but the same statement is not true for the same model under a different distribution. For this reason, the presence of BPs and the gradient-based trainability of a given parametrized learning model should always be discussed with respect to the same parameter distribution $\mathcal{P}$. In particular, it follows that, for the same distribution $\mathcal{P}$, a model that has a BP cannot be *efficiently* gradient-based trainable. In this direction, new research trends ask about specialized parameter distributions to improve the gradient-based trainability of variational QML models, under the name of *warm starts* (Puig-i-Valls et al., 2024).

As a closing remark, the importance of the initialization distribution is intuitive when discussing Neural Networks (NNs). Improvements in initialization have brought about major steps in the development of successful NN-based ML. This way, our definition of BPs for ML also captures the *vanishing gradients* problem of NNs (Hochreiter & Schmidhuber, 1997), strictly from the lens of the initialization distribution (and not from the lens of the activation function, normalization, regularization, or residual activations).

## F    PROOF OF THEOREM 3 AND COROLLARY 4

We first re-state and prove the results from Section 4.2, and next we briefly discuss their relation to the construction in Appendix B of Cerezo et al. (2023).

**Theorem 3** (Existence of trainable and non-dequantizable QML models.). *Let $\mathcal{X} = \{0,1\}^n$ and $\mathcal{Y} = \{0,1\}$. Let $Q(x)$ be a function in BQP and not in HeurBPP/poly under a given distribution $\mathcal{D}_{\mathcal{X}}$, and let $U(x)$ be a unitary such that $\langle 0|U^\dagger(x)Z_1U(x)|0\rangle = Q(x)$. Let $\mathcal{H}$ be a Hamiltonian, and $W(\vartheta)$ a parametrized unitary for which the following optimization problem can be solved with a given gradient-based algorithm $\mathcal{A}_W$:*

$$\vartheta^* \leftarrow \arg\max_{\vartheta \in \Theta} \langle 0|W^\dagger(\vartheta)HW(\vartheta)|0\rangle, \tag{F.1}$$

*and such that $\max_{\vartheta \in \Theta} \langle 0|W^\dagger(\vartheta)HW(\vartheta)|0\rangle = 1$. Call $V(x;\vartheta) = U(x) \otimes W(\vartheta)$, and $\mathcal{M} = Z_1 \otimes H$, and consider the corresponding hypothesis class $\mathcal{F}_Q$:*

$$\mathcal{F}_Q := \{f_\vartheta(x) = \langle 0|V^\dagger(x;\vartheta)\mathcal{M}V(x;\vartheta)|0\rangle \mid \vartheta \in \Theta\}. \tag{F.2}$$

*Let $\mathcal{D}$ specify a learning task: $\mathcal{D}(x,y) = \mathcal{D}_{\mathcal{X}}(x)\delta(y = Q(x))$.*

*Then $\mathcal{F}_Q$ is gradient-based trainable for $\mathcal{D}$, and it is not dequantizable.*

*Proof.* As explained in Gyurik & Dunjko (2023), for $Q : \{0, \dots, 2^n - 1\} \rightarrow \{0, 1\}$ to not be in HeurBPP/poly, $Q$ must be hard to evaluate and learn classically from examples. That means, that given polynomially many input-out pairs, no classical algorithm can evaluate it classically with probability larger than $1/2 + 1/\text{poly}(n)$ over $x \sim \mathcal{D}_{\mathcal{X}}(x)$. This could be for example the most significant digit of the discrete logarithm of $x$ according to a known basis. Based on $Q(x)$, we can consider a binary classification task in which each integer $x$ is assigned a class $y(x) \in \{\pm 1\}$ depending on $Q(x)$, by identifying the outcome measurement 0 with class $-1$.

In parallel, we have the optimization task of finding the highest energy of $H$ by optimizing the parameters of $V(\vartheta)$, which is assumed to be efficiently solvable.

Then as a supervised learning task, we consider a training set $S = \{(x_i, y_i)\}_{i=1}^N$, where $x_i \sim \mathcal{D}_{\mathcal{X}}(x)$ and $y_i = 1$ if $Q(x) = 1$, $y_i = -1$ if $Q(x) = 0$. The hypothesis class $\mathcal{F}_Q$ we consider is

$$\mathcal{F}_Q := \{f_\vartheta(x) = \langle 0|V^\dagger(x;\vartheta)\mathcal{M}V(x;\vartheta)|0\rangle \mid \vartheta \in \Theta\} \tag{F.3}$$

We consider the problem solved when we find a hypothesis specified by parameters $\vartheta_{\text{sol}}$ for which it holds that

$$f(x; \vartheta_{\text{sol}}) = y(x) \tag{F.4}$$

with high probability over $x \sim \mathcal{D}_{\mathcal{X}}(x)$. Using the training set we have access to, and using the mean squared error, we are left with the following optimization problem:

$$\vartheta_{\text{sol}} = \arg\min_\vartheta \{\frac{1}{2N}\sum_{i=1}^N \|f(x_i; \vartheta) - y_i\|^2\} \tag{F.5}$$

$$= \arg\min_\vartheta \{\frac{1}{2N}\sum_{i=1}^N \|\langle 0|V^\dagger(x_i;\vartheta)\mathcal{M}V(x_i;\vartheta)|0\rangle - y_i\|^2\} \tag{F.6}$$

$$= \arg\min_\vartheta \{\frac{1}{2N}\sum_{i=1}^N \|\langle 0|(U(x_i)\otimes W(\vartheta))^\dagger(Z_1\otimes H)(U(x_i)\otimes W(\vartheta))|0\rangle - y_i\|^2\} \tag{F.7}$$

$$= \arg\min_\vartheta \{\frac{1}{2N}\sum_{i=1}^N \|\langle 0|U(x_i)^\dagger Z_1 U(x_i)|0\rangle\langle 0|W(\vartheta)^\dagger HW(\vartheta)|0\rangle - y_i\|^2\} \tag{F.8}$$

$$= \arg\min_\vartheta \{\frac{1}{2N}\sum_{i=1}^N \|y_i\langle 0|W(\vartheta)^\dagger HW(\vartheta)|0\rangle - y_i\|^2\} \tag{F.9}$$

$$= \arg\min_\vartheta \{\frac{1}{2N}\sum_{i=1}^N \|y_i(\langle 0|W(\vartheta)^\dagger HW(\vartheta)|0\rangle - 1)\|^2\} \tag{F.10}$$

$$= \arg\min_\vartheta \{(\langle 0|W(\vartheta)^\dagger HW(\vartheta)|0\rangle - 1)^2\} \tag{F.11}$$

$$= \arg\max_\vartheta \{\langle 0|W(\vartheta)^\dagger HW(\vartheta)|0\rangle\} \tag{F.12}$$

$$= \vartheta^*. \tag{F.13}$$

We see that the solution of our classification problem $\vartheta_{\text{sol}}$ is the same as the solution $\vartheta^*$ of the optimization problem based only on $H$ and $W(\vartheta)$, which can be solved by the given training algorithm $\mathcal{A}_W$ by assumption. Because we assumed the optimization problem to be efficiently solvable by gradient-based optimization, it follows that our binary classification problem is also efficiently solvable by gradient-based optimization. Here, similarly to Prop. 1, we took a trainable model and added more ingredients to it to make it look difficult. But, in the end, training our model to fit data according to the function $Q$ is as easy as training the simple quantum state optimization task, so the model we proposed is gradient-based trainable. Still, the same task cannot be solved by a classical model because that would imply a classical algorithm being able to evaluate $Q(x)$, which we ruled out by assumption. $\qquad\square$

**Corollary 4** (Existence of trainable and non-dequantizable vari veryational QML model.). *There exist vari veryational QML models which are gradient-based trainable and non-dequantizable with any number of layers up to sub-exponentially many in the number of qubits.*

*Proof.* We prove this statement by giving an explicit example of a vari veryational model that fulfills the assumptions of Theorem 3. The model corresponds to a tensor product of two PQCs, one corresponding to the hard computational task of evaluating $Q(x)$, and one corresponding to the easy optimization task corresponding to $H$ and $W(\vartheta)$.

We start with the computational part, where we have the unitary $U(x)$ which implements the classically-hard function $Q(x)$. From $U(x)$, we must construct a layered circuit where all layers are equal, and which still produces $Q(x)$ when Pauli $Z$ is measured on the first qubit, at

the end. To achieve this, consider an auxiliary register made of $t \in \mathbb{N}$ qubits and consider an integer-adder $t$-qubit unitary gate $A$ whose action on the computational basis is to add 1 modulo $2^t$: $A|b\rangle = |b+1 \mod 2^t\rangle$. Then, we define the $(n+t)$-qubit unitary $\tilde{U}(x)$ as a sequential two-step process:

1. Implement $U(x)$ on the first $n$ qubits conditioned on the extra $t$ qubits being in state $|0\rangle$.

2. Apply $A$ on the $t$ working qubits.

With this, the action of $\tilde{U}(x)$ on the $(n+t)$ qubits is:

$$\tilde{U}(x)|0\rangle|b\rangle = \left(U(x)^{\delta_{b,o}}|0\rangle\right)|b+1 \mod 2^t\rangle. \tag{F.14}$$

Here $\delta_{a,b}$ is the Kronecker delta, which equals 1 if $a = b$, and is 0 otherwise. In particular, for the first $n$ qubits, $\tilde{U}(x)$ applies $U(x)$ if $b = 0$, and does nothing in any other case. Also, it follows that $\tilde{U}(x)^L|0\rangle|0\rangle = (U(x)^{\lceil L/(2^t)\rceil}|0\rangle)|L \mod 2^t\rangle$. And, in particular, $\tilde{U}(x)^L|0\rangle|0\rangle = (U(x)|0\rangle)|L\rangle$ for any $L < 2^t$. That means, when using $|0\rangle$ as the input state for both registers, applying $\tilde{U}(x)$ $L$ times in sequence has the same effect on the first $n$ qubits as only applying it once, given than $L$ is not exponential in $t$. So, provided the number of layers is subexponential $L < 2^t$, it holds that

$$\langle 0|\langle 0|(\tilde{U}^L(x))^\dagger(Z_1 \otimes \mathbb{I})\tilde{U}^L(x)|0\rangle|0\rangle = \langle 0|\langle 0|\tilde{U}(x)^\dagger(Z_1 \otimes \mathbb{I})\tilde{U}(x)|0\rangle|0\rangle \tag{F.15}$$

$$= \langle 0|U(x)^\dagger Z_1 U(x)|0\rangle \tag{F.16}$$

$$= Q(x). \tag{F.17}$$

For the easy optimization task, we could take any PQC-Hamiltonian combination that we know to be trainable and deep-layered. For this proof, it is enough to pick a very simple one. We can take the single-qubit observable $H = Z$, and the layered Ansatz $W(\vartheta) = \prod_{j=1}^L R_X(\vartheta_j)$. With these, it follows that initializing $\vartheta$ uniformly at random and then performing gradient descent is enough to solve the optimization task. Since the highest-energy state of $Z$ is already the initial state $|0\rangle$, this optimization problem reduces to finding a specification of $\theta$ fulfilling $\sum_{j=1}^L \theta_j = 2\pi M$, for any $M \in \mathbb{Z}$, since $R_X$ forms a $U(1)$ group and $R_X(2\pi M) = \mathbb{I}$ up to a global phase for any $M \in \mathbb{Z}$. This optimization problem is easy regardless of $L$, so in particular it also works for $L < 2^t$.

All together, the PQC we are considering consists of $L$ layers uniformly repeated:

$$\mathcal{F}_Q := \{f_\vartheta(x) = \langle 0|V^\dagger(x;\vartheta)\mathcal{M}V(x;\vartheta)|0\rangle \,|\, \vartheta \in \Theta\} \tag{F.18}$$

$$V(x;\vartheta) = \prod_{i=1}^L \tilde{U}(x) \otimes R_X(\vartheta_i). \tag{F.19}$$

We have seen that this PQC fulfills the assumptions of Theorem 3 and it also fulfills the defining requirements for a QML model to be vari veryational from Section 4.1. This completes the proof. $\square$

Both this construction and that of Cerezo et al. (2023) leverage a hard computational task to make a statement about trainability and dequantization. In both cases there is an encoding unitary which, upon measuring Pauli Z on the first qubit, evaluates a specific function of bitstrings to a single bit. The quantum function is chosen to ensure that no classical algorithm can evaluate it, even when given advice.

The main difference is two-fold. On the one hand, we pose a learning problem and a model that can solve it, and not just a computational problem. At the same time we also propose a learning algorithm that ensures that the quantum algorithm actually solves the task. On the other hand, we force ourselves to work with so-defined vari veryational models, which rely on deep-layered variational circuits, as opposed to the circuit constructed in Cerezo et al. (2023). That being said, a valid criticism of our construction is that the way in which we turn a computational task into a learning task is almost trivial. The trainable component of our circuit is detached from the encoding part, and it only appears as a multiplication factor independent of the input. Indeed, our proof does not educate us on how to design PQCs for QML, but rather it is the minimal construction necessary to prove the statement.

