# OpenReview forum: "On the Relation between Trainability and Dequantization of Variational Quantum Learning Models"
_ICLR.cc/2025/Conference — ICLR 2025 Poster_

### Official Review · Reviewer_LzqN · 2024-10-24

**Soundness:** 4
**Presentation:** 4
**Contribution:** 3
**Rating:** 8
**Confidence:** 5

**Summary:**

The authors address a pressing question in the field of quantum machine learning: do there exist "variational" (i.e., trained via gradient descent) quantum neural networks which are both efficient to train and also difficult to classically simulate? The authors answer this question in the affirmative, giving a general method for constructing variational quantum machine learning models with these two properties.

**Strengths:**

The authors prove concrete, rigorous results, which are hard to come by in the field of quantum machine learning; this is particularly true of concrete quantum-classical separations in machine learning (where the quantum model is assumed to be efficiently trainable). The authors give explicit examples of efficiently trainable quantum networks with provable quantum advantages in learning over their classical counterparts.

**Weaknesses:**

The authors' Theorem 3 and Corollary 4---which give the recipe for constructing a learning problem solved by a quantum model efficiently trainable via gradient descent, but which is unable to be dequantized---essentially separates these two conditions out, considering a tensor product of two systems where one is trivial and easy to optimize and the other implements a quantumly-easy, classically-hard function, but is untrained. Because of this, it is difficult to know what broader impact this work has on the actual design of variational quantum algorithms (see Questions).

A minor additional weakness is that the authors' proposed criteria for "dequantizable" focuses on the supervised learning setting; the unsupervised learning setting (e.g., sampling problems) may also contain concrete quantum advantages in an efficiently trainable setting, particularly with the current wealth of quantum experiments demonstrating a quantum advantage in sampling tasks (Nature 574, 505; Nature 626, 58).

**Questions:**

What implications do the authors' results have on the construction of quantum neural network architectures?

---

> ### Author Response · Authors · 2024-11-16
> **Response to Summary, Strengths, and Weaknesses**
>
> We thank the reviewer for their positive and helpful assessment of our work.
> We believe this summary correctly identifies our core contributions in terms of resolving a pressing question and providing a general method for constructing variational QML models.
> It makes us very happy that the reviewer highlights the rigor of our results, which was a concrete goal for us while preparing this submission.
> In particular, we are pleased to contribute to the corpus of quantum separations in learning with variational QML models.
>
> > The authors' Theorem 3 and Corollary 4---which give the recipe for constructing a learning problem solved by a quantum model efficiently trainable via gradient descent, but which is unable to be dequantized---essentially separates these two conditions out, considering a tensor product of two systems where one is trivial and easy to optimize and the other implements a quantumly-easy, classically-hard function, but is untrained.
> > Because of this, it is difficult to know what broader impact this work has on the actual design of variational quantum algorithms (see Questions).
>
> We agree with the reviewer that this was one of the least clear points in our original submission, as it was also raised by reviewers K1hL and to some extent 1RRV.
> We have added new Sections 5 and 6 to better discuss precisely the breadth and depth of impact of our work.
> For the actual design of variational quantum algorithms, we have avoided relating our construction to other pre-existing trends, largely because of the dire perspectives put forward in Cerezo et al. (2023).
> The current broadly-accepted opinion is that all generic variational QML models to date can be either trainable or non-dequantizable, but not both.
> For this reason, we concentrate on showing that there exist generic variational QML models that fulfill both those properties.
> One message to be drawn is: we likely need to rethink the design principles we had until now, since they result in an alignment between trainability and dequantizability.
> Another message, more direct, is: following our prescription we are guaranteed to check two important boxes: trainability and non-dequantization.
> This should inspire the community to look for other novel schemes that achieve comparable scores also for expressivity and generalization when compared to current approaches suffering from dequantization.
> Future work is needed to improve on our constructions, but our paper proposes footing from where to start building toward actual practical relevance.
>
> > A minor additional weakness is that the authors' proposed criteria for "dequantizable" focuses on the supervised learning setting; the unsupervised learning setting (e.g., sampling problems) may also contain concrete quantum advantages in an efficiently trainable setting, particularly with the current wealth of quantum experiments demonstrating a quantum advantage in sampling tasks (Nature 574, 505; Nature 626, 58).
>
> We thank the reviewer for pointing this missing gap in the literature, we agree we should have mentioned this line of work in our original submission, and so we have added a paragraph in Section 2.4 commenting on trainability and dequantization of unsupervised learning, as well as a short sentence in the first paragraph of the new Section 5.
> The random-circuit-sampling problem has received a fair share of attention and has been formalized in Sweke et al. (2021).
> We then consciously chose to stay in the domain of supervised learning.
> It is still clear to us that we should mention unsupervised learning, for clarity and completeness.
>
> M. Cerezo et al. *Does provable absence of barren plateaus imply classical simulability? or, why we need to rethink variational quantum computing*. arXiv:2312.09121, (2023)
>
> F. Arute et al. *Quantum supremacy using a programmable superconducting processor*. Nature, **574**, 505, (2019)
>
> D. Bluvstein et al. *Logical quantum processor based on reconfigurable atom arrays*. Nature, **626**, 58, (2023)
>
> R Sweke et al. *On the quantum versus classical learnability of discrete distributions*. Quantum, **5**, 417, (2021).

---

> > ### Comment · Reviewer_LzqN · 2024-11-20
> >
> > We thank the authors for their updating of the work to make it more clear, including mentioning the unsupervised learning setting. However, we would like more clarification in the authors' response to our question.
> >
> > We agree with the authors that Cerezo et al. (2023) broadly ruled out "natural" variational QML models which are trainable and non-dequantizable, as discussed in the authors' new Sec. 5. In their response to our initial question, the authors claim that their construction provides "footing from where to start building toward actual practical relevance." To rephrase what was the original question: broadly, how _should_ one construct quantum machine learning models so that they have a quantum advantage and are efficient to train? In other words, what is precisely the "footing" from which to build? Is there a setting where the authors' construction given in the proof of Proposition 1 would be used, or a way in which it is "able to generalize to other QML designs" (as put by Reviewer K1hL)?

---

> > > ### Author Response · Authors · 2024-11-21
> > > **Further clarification on footing from where to start building toward actual practical relevance.**
> > >
> > > We would like to thank the reviewer for their kind words and appreciation of the improvements to the manuscript. We apologize if we seem to have dodged the original meaning of the reviewer’s question, this was not intended. We believe this question is really at the core of our contribution, and we appreciate the chance to further elaborate. This work is the result of the authors wondering at length what should be on the wishlist for “good QML models”. We may not have transmitted this idea better, but one issue is to compile such a list in a precise way.
> > >
> > > If we take the words of the reviewer literally, in order to construct (variational) QML models that have a quantum advantage and are efficient to train, one should only follow the construction we give in Theorem 3 and Corollary 4. These results provide a recipe for precisely *quantum advantage* and *efficient to train*, and nothing else. We are aware that we should not take the words of the reviewer literally, there must be something else for us to add to the wishlist. One indirect contribution of our work is to point out that “just” demanding trainability and non-dequantizability is manifestly not enough, as it allows for practically uninteresting constructions. This induces a tension, because all natural-looking QML models proposed until now are either trainable or non-dequantizable, but not both. To us, this means we should *not* be trying to generalize to other QML designs, since they do not even fulfill the *clearly insufficient* wishlist we start from.
> > >
> > > The footing we provide in moving forward is to introduce non-dequantization as a design feature for future QML models. We explain how one can near-generically construct Ansätze for which at least dequantization is not an issue, as we make sure the Ansätze also have a computational role. Said otherwise: we start with an incomplete wishlist of necessary properties, we realize that most-if-not-all existing models fail to qualify for our wishlist, and we finally prove that the set of QML models that fulfill the wishlist is not the empty set. After this, we also realize that the constructions we use for our proofs are still a way away from answering the ultimate main question in the field. We agree, more work is needed. To us, this means our contribution is at least clearing out this situation.
> > >
> > > The recent work of Deshpande et al. (2024) also designs variational QML models wielding “classical hardness” as a design feature. In Deshpande et al. (2024), the authors propose one specific way of parametrizing a set of reachable functions using intermediate measurements and feed-forward operations. It follows that while the footing we provide is not a panacea, it provides intuition to other QML researchers on what kind of hard computations can be illuminating about what is meant to be learned, and then use quantum circuits to explore the space “around” these hard computations. We must still make ourselves more familiar with the newly proposed architecture. For the moment we have no claims about dequantization, as similar constructions have been previously deemed average-case classically simulable, while being worst-case classically intractable.
> > >
> > > Trying to answer the more general question “how should one construct practically relevant QML models?”, our approach would be: to add sensible formal properties to the wishlist. We do not shy away from the fact that critical work is needed for our work’s contributions to be completely fulfilled, and we ourselves are already researching in that direction. Specifically about the construction in the proof of Proposition 1 mentioned in the reviewer’s question: our construction consists of “compiling” a kernel-based QML model as a variational QML model. The training algorithm we propose gives the same output as the kernel method would. For this reason, whether one should use the variational construction or the original quantum kernel model would come down to hardware considerations only. In principle the quantum circuits involved in both alternatives are the same, though our construction comes with a compilation as a hardware efficient ansatz. We suspect this will not be a make-or-break consideration, as our construction served mostly a theoretical purpose: to show that Barren Plateaus and trainability are essentially different concepts.
> > >
> > > Deshpande et al. *Dynamic parameterized quantum circuits: expressive and barren-plateau free*, arXiv: 2411.05760 (2024)

---

> > > > ### Comment · Reviewer_LzqN · 2024-11-22
> > > >
> > > > We thank the authors for their detailed response; the relation to kernel-based QML models is interesting. We have raised our score in response.

---

> > > > > ### Author Response · Authors · 2024-11-24
> > > > > **Thank you for the review**
> > > > >
> > > > > We thank the reviewer for their effort in helping us improve our manuscript, and for updating their evaluation.

---

> ### Author Response · Authors · 2024-11-16
> **Response to Questions**
>
> > What implications do the authors' results have on the construction of quantum neural network architectures?
>
> We presumably have already answered this question in our response to the Weaknesses.
> But briefly: we first remark that current design principles for the construction of variational QML models seem to be doomed, and based on this a clear implication is the possibility of using “classical hardness” as an explicit feature in the design of variational QML models.
> We recognize that it will require future work to design variational QML models that have a chance of being practically relevant, but we believe we offer a solid place from where to start.

---

> ### Author Response · Authors · 2024-11-16
> **Overall assessment**
>
> We appreciate how reviewer LzqN highlighted the explanations of broader impact and missing literature links as areas of potential improvement.
> Prompted by these, we have added clarification text in Sections 2, 5, and 6.
> We believe these changes improve the overall quality of the manuscript, and hope the reviewer also sees it this way when evaluating our updated submission.

---

### Official Review · Reviewer_sYUX · 2024-10-29

**Soundness:** 3
**Presentation:** 3
**Contribution:** 2
**Rating:** 8
**Confidence:** 3

**Summary:**

This paper formalizes the relationship between differing ideas in the QML literature regarding different aspects of learning QML models. With these formalizations in hand, the authors then prove that there exist variational QML models which are trainable and non-dequantizable.

**Strengths:**

- Figures 1 and 3 are good, and help with reader comprehension
- The paper is generally well written and understandable
- The writing style especially is very approachable and the focus on explanation is beneficial to the paper
- The coverage of recent literature is sufficient

**Weaknesses:**

- This paper seems to want to do two things. After introducing a new representational/formal scheme for QML, it wants to (a) show how other papers fit into this scheme and (b) use this scheme to prove novel results. However, in attempting to do both, results in both being weaker. If the paper were to lean into (a) and be more of a review paper (where the formalize in the synthesis of many previous papers) and give more examples of how the literature fits into this paradigm, it would be good. Or, if the paper were to lean into (b) and emphasize more the research power of this paradigm through novel theoretical contributions, that would also be good. To be concrete for this latter approach, this would largely involve moving 3.3/3.4 to an appendix.
- Figure 2 and Table 1 seem to convey the same information redudantly
- This paper is generally interesting and is a quality paper, but I question whether ICLR is the ideal venue for it and whether there is sufficient novelty of interest to this community
- Not a major point, but the “vari veryational” term doesn’t seem ideal, both for speaking (sorry for the phonocentrism Derrida), but also for writing (with autocorrect/spell check). Maybe a different abbreviation would clarify better?

**Questions:**

- If “de-” is the prefix for the reverse of something, can non-dequantizable just be “quantizable”?

---

> ### Author Response · Authors · 2024-11-16
> **Response to Summary and Strengths**
>
> We value the effort on the reviewer’s side to carefully understand our work and help us enhance it.
> We believe this summary accurately represents the merits and contributions of our submission.
> We enjoy the positive feedback on the overall clarity and accessibility of our work.
> We are happy to hear that the effort we put into writing an approachable text has been appreciated by the reviewer.
> We thank the reviewer for highlighting the sufficiency of our literature coverage.
> We tried to be comprehensive, but it is perfectly possible that we missed some relevant works, which we would be happy to include if the reviewer has some in mind.

---

> ### Author Response · Authors · 2024-11-16
> **Response to Weaknesses (1/2)**
>
> > This paper seems to want to do two things.
> > After introducing a new representational/formal scheme for QML, it wants to (a) show how other papers fit into this scheme and (b) use this scheme to prove novel results.
> > However, in attempting to do both, results in both being weaker.
> > If the paper were to lean into (a) and be more of a review paper (where the formalize in the synthesis of many previous papers) and give more examples of how the literature fits into this paradigm, it would be good.
> > Or, if the paper were to lean into (b) and emphasize more the research power of this paradigm through novel theoretical contributions, that would also be good.
> > To be concrete for this latter approach, this would largely involve moving 3.3/3.4 to an appendix.
>
> The reviewer identified an issue that we indeed struggled with in the conversations about how to present our work, and we’re thankful for the provided valuable advice.
> We agree with the assessment of the reviewer, and we followed their advice by moving Sections 3.3 and 3.4 into the new Appendix C.
> We have decided against further thinning the parts of the text that would fall under category (a) because we believe that the context in which our work arises should not be understated.
> A critical question had been floating in the air that challenged the viability of the entire field by half-conjecturing that there could be no variational QML models that are simultaneously trainable and non-dequantizable.
> Except the language in which this question had been stated was that of variational optimization problems, and not QML.
> Further, formal language was generally missing that allowed for a precise study of the problematic.
> These are the reasons why we would be reluctant to remove much more content from Section 3, but we are eager to hear the reviewer’s thoughts.
>
> > Figure 2 and Table 1 seem to convey the same information redundantly.
>
> Thank you for catching this, this was an artifact from a previous version of this manuscript.
> We have removed Table 1.
>
> > Not a major point, but the “vari veryational” term doesn’t seem ideal, both for speaking (sorry for the phonocentrism Derrida), but also for writing (with autocorrect/spell check).
> > Maybe a different abbreviation would clarify better?
>
> We thank the reviewer for raising this point, as it has been a point of tension for the authors.
> The main advantage of “vari veryational” is that it perfectly captures the concept we want to transmit: that these models are almost very variational, but not quite.
> The current option of “vari veryational” won mostly due to the peculiar sense of humor the authors seem to share (and, we agree, not for purely scientific reasons).
> By taking this humorous option, we anticipate that this term will not be used extensively.
> Instead, we hope it brings attention to the fourth paragraph of Section 4.1, where we extend an invitation to the community to help us converge to a good definition of very variational.
> In the future, either we have an agreed-upon definition of very variational, or these ideas have been forgotten, but in both cases the need to speak up or type down “vari veryational” vanishes quickly.
>
> That being said, we have spent more time considering alternatives, also prompted by a question by reviewer K1hL.
> We are ready to change the name if we find a better one, but until now everything we could think of fails at transmitting the core message: that these models are almost very variational, but not quite.
> Some of the alternatives are: quasi-variational, semi-variational, almost-variational, para-variational, and similar.
> But all of these fail on the account that these models are variational, just not very variational.

---

> ### Author Response · Authors · 2024-11-16
> **Response to Weaknesses (2/2)**
>
> > This paper is generally interesting and is a quality paper, but I question whether ICLR is the ideal venue for it and whether there is sufficient novelty of interest to this community.
>
> We appreciate the kind words by the reviewer, we take “generally interesting” and “quality paper” as notable praise.
> This weakness is similar to the first weakness raised by reviewer 1RRV’s, which makes us further convinced that we should have done a better job at clarifying the interest and novelty of our work.
> By streamlining parts of Sections 2 and 3 as advised by the reviewers, we hope to bring our results more front and center, and we have added new discussion in Section 5 to consider our contributions under complementary perspectives.
>
> We would like to take a chance to explain why we decided to submit this work to ICLR based on what we believe are the novelty and potential impact of our work.
> We apologize for partially repeating the response to reviewer 1RRV, but upon reading the submission and review guidelines, we understood that the scope and range of ICLR are papers reporting novel findings and maintaining a high degree of correctness and scientific rigor.
> We chose to submit our work to ICLR because we are convinced that our work presents novel findings that are broadly and deeply relevant to the QML community.
> We believe the ultimate goal in the field of QML is to find a quantum advantage in learning using variational QML models for a real-life, practically relevant task.
> Recent progress in this direction has been called into question by the threat of “trainability implies dequantization”.
> According to well-spread opinions presented in Cerezo et al. (2023), the situation is dire: most-if-not-all generic variational QML models proposed to date can be either trainable or non-dequantizable, but not both.
> If no variational QML model exists that is trainable and non-dequantizable, then there is no hope for quantum advantage, and we cannot hope to eventually reach practical relevance.
> From this it follows that the road towards practical relevance is most likely not following current trends in the design of variational QML models.
> The analytic tools used to characterize this situation resulted in a seemingly iron-clad case against the existence of generic variational QML models being trainable and non-dequantizable.
> The picture resulting from the study of polynomial-sized Dynamical Lie Algebras (DLAs) does not offer much room for escape.
>
> Recognizing this state of affairs, we establish formal language to not only clear the current situation, but also to help us move forward as a field toward practical relevance.
> At the same time, our main technical contribution is the resolution of an important open question: we prove a rigorous quantum advantage in learning using variational QML models; we further propose a general recipe to yield domain-specific advantageous variational QML models for any quantum function that is classically intractable, as we highlight in the new paragraph in Section 6, we propose “the ability to evaluate classically intractable functions” as a design feature.
> This convinces us that our work is novel at least on two accounts: we prove a solid result that had eluded the community until now (made possible by our formalization of the key concepts), and we offer this formalization to move forward toward practical relevance.
> In terms of impact, our contributions have far-reaching consequences: by giving a general recipe our results can be further specialized to a broad spectrum of domains of application (as briefly discussed in the new Section 5), and by realizing that existing trends are unlikely to yield practical relevance unless major breakthroughs are found, our contribution has the potential to deeply shift the paradigm on how we approach the design of variational QML models.
>
> M. Cerezo et al. *Does provable absence of barren plateaus imply classical simulability? or, why we need to rethink variational quantum computing*. arXiv:2312.09121, (2023)

---

> ### Author Response · Authors · 2024-11-16
> **Response to Questions**
>
> > If “de-” is the prefix for the reverse of something, can non-dequantizable just be “quantizable”?
>
> This is a good question, and it highlights the issue of having to say “non-” at every occasion.
> Unfortunately, we believe there is no alternative to the current formulation.
> Our assessment may be a bit subjective, but we would say that “quantizable” is the property that something “can be quantized”.
> Since this property lies at the very core of quantum mechanics, “quantizable” already carries historic meaning that likely make it confusing if we used it to mean “quantum machine learning models that cannot be de-quantized”.
> This is because “quantizing” is a partial synonym of “discretizing”, while “de-quantizing” has become a synonym of “making something classical”.
> Since “dequantization” in our sense is essentially a one-way process, there is no standard word for “the impossibility of this process”, and simply removing the prefix “de-” linguistically would go in the direction of “reverting the process”, which is not what we want to capture.
>
> An alternative could be “robust against dequantization”, though this would likely prove too long to be a convenient replacement for “non-dequantizable”.

---

> ### Author Response · Authors · 2024-11-16
> **Overall assessment**
>
> We thank the reviewer for providing concrete feedback on improving the presentation of our text.
> Prompted by the question regarding ICLR being the correct venue for our work, we have improved the explanation of the nature and impact of our contributions in the new Sections 5 and 6.
> We are very satisfied with the updated streamlined version of our submission, and we hope the reviewer assesses positively the improvements they helped accomplish in the text.

---

> > ### Comment · Reviewer_sYUX · 2024-11-23
> >
> > I appreciate the thoroughness of the authors responses. I believe their changes will improve the paper and I have adjusted my score to reflect that. I also accept that ICLR is an acceptable venue for this work.
> >
> > > The reviewer identified an issue that we indeed struggled with in the conversations about how to present our work, and we’re thankful for the provided valuable advice. We agree with the assessment of the reviewer, and we followed their advice by moving Sections 3.3 and 3.4 into the new Appendix C. We have decided against further thinning the parts of the text that would fall under category (a) because we believe that the context in which our work arises should not be understated. A critical question had been floating in the air that challenged the viability of the entire field by half-conjecturing that there could be no variational QML models that are simultaneously trainable and non-dequantizable. Except the language in which this question had been stated was that of variational optimization problems, and not QML. Further, formal language was generally missing that allowed for a precise study of the problematic. These are the reasons why we would be reluctant to remove much more content from Section 3, but we are eager to hear the reviewer’s thoughts.
> >
> > I think moving 3.3/3.4 is sufficient, just to help streamline the narrative I wouldn't ask for any more text removals.

---

> > > ### Author Response · Authors · 2024-11-24
> > > **Thank you for the review**
> > >
> > > We thank you for the focused and targeted comments on our manuscript, and for positively evaluating the changes we have implemented.

---

### Official Review · Reviewer_1RRV · 2024-11-01

**Soundness:** 2
**Presentation:** 3
**Contribution:** 2
**Rating:** 6
**Confidence:** 4

**Summary:**

This paper tries to establish a relationship between trainability and dequantization in the context of QML models. The authors present a formalization of these concepts and attempt to illustrate conditions under which these properties co-exist.

**Strengths:**

- The paper commendably formalizes the concept of dequantization, linking several key concepts in QML models, including trainability, dequantization, and classical simulation. This integration provides a valuable framework for understanding QML models.

- The paper is well written and concepts are explained clearly.

**Weaknesses:**

- From a technical standpoint, the paper's contribution appears limited, primarily synthesizing existing results rather than offering new findings. It attempts to establish connections between different unclear concepts but lacks significant technical contributions.

- The discussion would benefit from a more detailed analysis of existing QML models to determine which categories they fall into. Such a comparison would enhance the paper's relevance and applicability in the field.

- On the practical side, the paper falls short of providing actionable insights or guidelines for designing effective QML models, which limits its utility for practitioners in the field.

**Questions:**

- It would enhance the paper if the authors could offer insights or preliminary guidelines on designing QML models that are trainable, non-dequantizable, and very variational.

---

> ### Author Response · Authors · 2024-11-16
> **Response to Summary and Strenghts**
>
> We thank the reviewer for their time and interest in improving our submission.
> We appreciate the honest criticism as it gives us a way to further clarify and highlight our contributions.
> We feel there may be a misalignment between the merits of our work and the qualities for which it has been judged in this review, so we would like to take the chance to hopefully clarify our contributions, both conceptual and technical.
>
> We feel somewhat uneasy that the reviewer considers that our work only “tries to establish” a relationship between trainability and dequantization, and that our formalization merely “attempts to illustrate” conditions under which these properties co-exist.
> It is our view that we do establish the relationship, and that our formalization does precisely illustrate the conditions, among other contributions.
> Said otherwise, the statements in this summary are all correct - and it seems to us that we agree with the reviewer on these, but in our view they may have under-represented the novelty and impact of our submission.
> We wonder whether the depth of our results has not been appreciated, which would also explain the low soundness score, while no specific problems with the soundness of our work are highlighted in the review.
>
> Again, we take this whole-heartedly as a chance to improve on our work. We will be more explicit on our contributions in the remainder of the rebuttal, and we have better highlighted our contributions by separating the Discussion (new Section 5) from the Conclusion (new Section 6) in the revised manuscript.
>
> We thank the reviewer for their positive feedback on our formalization, as well as our work in linking relevant concepts in QML.
> We appreciate the notion that our work is a valuable contribution.
> We may be optimistic that “understanding QML models” includes a broad range of useful applications, thus potentially highlighting the potential impact of our work.

---

> ### Author Response · Authors · 2024-11-16
> **Response to Weaknesses (1/2)**
>
> > From a technical standpoint, the paper's contribution appears limited, primarily synthesizing existing results rather than offering new findings.
> > It attempts to establish connections between different unclear concepts but lacks significant technical contributions.
>
> We appreciate the critical feedback from the reviewer.
> We respectfully disagree on what are the core contributions of our work.
> While we do connect some concepts, this is not the main point of the work nor the majority of the contribution in our view.
> Instead we formalize the necessary criteria needed to eventually make solid progress in the field.
> Our goal is to be precise, not to re-state existing results.
> Further, by delving into our formalism, we are able to prove important rigorous results that had remained elusive until now.
> In this sense, we also believe these are indeed new findings, and moreover key findings for a large and broadly recognized debate in the QML community.
>
> We can understand the view of the referee: the immediate consequences of our results are not a list of design principles for QML that can rival the performance of well-established classical ML, which had been the perhaps-overly-enthusiastic dream in the early days of the field.
> In fact, we agree with the reviewer that our core contributions are conceptual in nature, more than technical.
> But then again - we do formally prove the existence of new classes of models in a constructive way, so it is also not just conceptual.
> Furthermore, we have a differing opinion about our work lacking significant technical contributions, as we prove a rigorous generic quantum advantage in supervised learning, of which there is no abundance in the literature.
>
> In our search for the right venues for our work, we analyzed the scope and range of ICLR’s call for papers.
> The reviewer guide lists as strong points: “technical correctness, [...], and novel findings” among others.
> It is our understanding that we should be judged on the genuine interest and novelty of our findings, and not exclusively on the technical aspects of our contributions.
> Indeed, as also listed under considerations for reviewing submissions: the “objective of the work” is mainly conceptual, and this is fully aligned with the stated scope of this conference.
> For this reason we believed, and still believe, that ICLR is the right venue for our work, as we do provide novel findings that are broadly and deeply relevant to the QML community.
>
> Nonetheless, the concerns of the reviewer are fully understandable, and while we don’t think they fully apply to our work, we see them as guidelines on how to improve the presentation and content.
> In this direction, we have replaced what used to be Section 5 with two new sections: Discussion (new Section 5) and Conclusion (new Section 6).
> We have added several paragraphs to hopefully better highlight the nature and depth of our contribution versus the known results before our work.
> To avoid confusion, we have also moved part of the related work section into an appendix: what used to be sections 3.3 and 3.4 are now mostly the new Appendix C.

---

> ### Author Response · Authors · 2024-11-16
> **Response to Weaknesses (2/2)**
>
> > The discussion would benefit from a more detailed analysis of existing QML models to determine which categories they fall into.
> > Such a comparison would enhance the paper's relevance and applicability in the field.
>
> We have extended this part of the related work section.
> Following and earlier comment and also a comment from reviewer sYUX, though, we have deferred most of this discussion to the new Appendix C.
>
> > On the practical side, the paper falls short of providing actionable insights or guidelines for designing effective QML models, which limits its utility for practitioners in the field.
>
> As mentioned earlier, and as stated in Section 5 of the original submission, we agree that our results don’t immediately provide guidelines for designing effective QML models.
> To our knowledge, there are precious few (if any) such actionable insights across the entire literature of QML, and none that do not immediately suffer from dequantization or expressivity, so we would not count this as a major weakness of our work, nor one that could have been reasonably expected for us to fill.
>
> A simple and potentially helpful perspective is to distinguish current research between “how to QML?” and “why to QML?”.
> In this picture, we would place ourselves firmly in the latter (although not exclusively).
> We would also understand that providing actionable advice should be the manifest goal of works that fall in the first category.
> We fundamentally agree with the reviewer that our ultimate goal is to understand how to design good QML models, which can eventually rival classical ML methods on practically relevant tasks.
> As we argue in the introduction and discussion, though, we identified that a crucial previous question was still open.
>
> We venture the possibility that the reviewer has judged our work as “how to QML?”, and we would like to respectfully request that it be judged as “why QML?”. We hope our new Sections 5 and 6 help bring forward these nuanced points and better explicate the novelty and nature of our contributions.

---

> ### Author Response · Authors · 2024-11-16
> **Response to Questions**
>
> > It would enhance the paper if the authors could offer insights or preliminary guidelines on designing QML models that are trainable, non-dequantizable, and very variational.
>
> We thoroughly agree that “offering insights or preliminary guidelines on designing QML models that are trainable, non-dequantizable, and very variational” would enhance the quality of the paper.
> To our understanding, though, this is precisely what our paper does!
> Except the reviewer carefully wrote “very variational”, and manifestly not “vari veryational”.
> This is very much in line with the issue that “variationalness” is not a well-defined quantity.
> In Section 4.1 we introduce “vari veryational” QML models, and in the fourth paragraph we extend an invitation to the community to converge into a good definition of very variational.
> The current status is that there is no proposed attempt at formalizing “very variational” QML models beyond our “vari veryational” ones.
> We are excited for the community’s proposals to improve on the list of structural properties.
> But we would argue our definition “vari veryational” is reasonably acceptable, and then we would claim that Theorem 3 and Corollary 4 provide precisely what the reviewer requests.
> These results are, precisely, “insights and preliminary guidelines on designing QML models that are trainable, non-dequantizable, and vari veryational”.
>
> We have further added a  last paragraph in the new Section 6 address some of these points that we failed to make clearer in our original submission.

---

> ### Author Response · Authors · 2024-11-16
> **Overall assessment**
>
> We thank the reviewer once again for their valuable feedback on our work.
> Prompted by the well-targeted comments, we have extensively clarified the nature and reach of our contributions in the new Sections 5 and 6.
> We have also added a discussion on how existing QML models fit in our categories in the new Appendix C.
> With these changes together with the comprehensive explanation in this response, we hope the reviewer shares a positive evaluation of our updated manuscript with respect to the original submission.

---

> ### Comment · Reviewer_1RRV · 2024-11-26
>
> Thanks for the detailed response. I have no further questions and also raised the score.

---

> > ### Author Response · Authors · 2024-11-27
> > **Thanking reviewer 1RRV**
> >
> > We appreciate the effort on the reviewer's side in improving our manuscript, and their positive assessment of the modifications prompted during the review process.

---

### Official Review · Reviewer_K1hL · 2024-11-04

**Soundness:** 3
**Presentation:** 3
**Contribution:** 3
**Rating:** 6
**Confidence:** 3

**Summary:**

This paper discusses and clarifies several important concepts in quantum machine learning (QML), including trainability, simulatability, and de-quantization. Building upon the established concepts, this paper introduces a new family of "vari veryational" QML models, a spoonerism of "very variational", that capture the essence of deep-layered QML models and gradient-based training algorithms. It is shown that in this family of QML models, there exist non-dequantizable but trainable models. This resolves an open question that trainability and dequantization are mutually compatible.

**Strengths:**

- The paper proposes clear definitions for trainability and dequantization using a rigorous learning theory language. This clarifies the vagueness of many seemingly related but not equivalent concepts in quantum learning theory.
- This paper constructed a QML model that is gradient-based trainable but not dequantizable (based on standard cryptographic assumptions).
- This paper provides an extended discussion of several related results in the quantum learning theory (Figure 3).

**Weaknesses:**

- The QML model constructed in this paper seems a bit contrived. The construction is based on a computationally hard problem, and the proposed training method is quite specific and not able to generalize to other QML designs. I feel this construction is mostly of theoretical interest and has limited connection to practically relevant variational quantum algorithms.
- Several definitions in Section 2 are quite formal and math-heavy, and it’s unclear whether such definitions are truly necessary in light of the paper’s main technical contributions. I feel the theoretical framework may be somewhat excessive relative to the provable results.

**Questions:**

- What does "gradient-based trainable" mean precisely? Does it mean there is no barren plateau in the sense of Definition 2?
- On page 2, the risk functional is defined over the space $\mathcal{Y}^\mathcal{X}$. Can you explain this notation, specifically, why the data domain $\mathcal{X}$ appears as the power in the label co-domain $\mathcal{Y}$, but not vice versa?
- I am a bit uncertain about the name "vari veryational". Is there further justification for this name, besides it's a spoonerism of "very variational"? It feels a bit uninformative to the general quantum information audience.

---

> ### Author Response · Authors · 2024-11-16
> **Response to Summary, Strenghts, and Weaknesses**
>
> We appreciate the reviewer’s efforts to improve our submission.
> We thank the reviewer for the detailed work and thoughtful reading and review.
> We celebrate that the clarity and rigor of our work are highlighted as main strengths.
> Indeed, one of the goals of our work was to serve as a basis for further discussion.
>
> > The QML model constructed in this paper seems a bit contrived.
> > The construction is based on a computationally hard problem, and the proposed training method is quite specific and not able to generalize to other QML designs.
> > I feel this construction is mostly of theoretical interest and has limited connection to practically relevant variational quantum algorithms.
>
> We agree with the reviewer that the models constructed in this paper can be considered contrived, we highlighted this issue in the third paragraph of Section 5 in the original submission.
> Since our objective was to resolve a pressing open question about trainability and dequantization, indeed our results are theoretical, but we do believe they have critical practical implications.
> As we explain in an added paragraph at the end of the new Section 6: our work arises in a context where the entire field of variational QML has been called into question, where doubt has been cast on the possibility of variational QML models being both trainable and dequantizable.
> Critically, progress toward useful variational QML models necessitates these doubts be cleared.
> We notice that in recent discussions of trainability versus dequantizability (where these terms had not been properly defined), the fact that quantum computers can compute hard things tends to be ignored, and so we construct models based on this feature.
> We provide a broadly applicable recipe for designing variational QML models that are trainable and non-dequantizable, as we now remark in the third paragraph of the new Section 5.
>
> Just as a side remark: we agree that the models constructed in our work may be contrived, but we disagree that “the proposed training method is quite specific and not able to generalize to other QML designs”, as the proposed training methods are the broadly-applicable gradient descent and kernel-based training.
> Proving quantum advantage in learning with arbitrary training algorithms is easier than proving quantum advantage in learning with everyday-looking training algorithms, and in our work we provide the latter.
>
> The reviewer notes that our contribution has limited connection to practically relevant variational quantum algorithms.
> In the specific context of QML, we have a different opinion: the community generally agrees with Cerezo et al. (2023) that there are some fundamental issues in the way QML models have been designed until now.
> There is to date no convincing evidence that any of the variational QML models proposed to date are practically relevant.
> We would like to highlight that in our work we are not trying to establish connections to other variational QML models, rather we aim to fix the language that will enable us to achieve real practical relevance.
> It has always been our goal to reach practical relevance, but by now it has become clear that the way to practical relevance is not following the mainstream trends in the field.
> Our added text in the new Sections 5 and 6 aims to clarify some of these points.
>
> For these reasons, we do believe that while our work is predominantly theoretical in nature, it does have non-trivial practical repercussions.
>
> > Several definitions in Section 2 are quite formal and math-heavy, and it’s unclear whether such definitions are truly necessary in light of the paper’s main technical contributions.
> > I feel the theoretical framework may be somewhat excessive relative to the provable results.
>
> We thank the referee for pointing out this imbalance between the level of formalism in the definitions and in the results.
> Indeed, we noticed that the first part of Section 2 contained much technical language that was not used later on.
> Accordingly, we streamlined this first part of Section 2, removing unnecessary notation, and concentrating only on the elements of supervised learning that we use in our definitions and results.
>
> M. Cerezo et al. *Does provable absence of barren plateaus imply classical simulability? or, why we need to rethink variational quantum
> computing*. arXiv:2312.09121, (2023)

---

> > ### Author Response · Authors · 2024-11-16
> > **Response to Questions**
> >
> > > What does "gradient-based trainable" mean precisely?
> > > Does it mean there is no barren plateau in the sense of Definition 2?
> >
> > We recognize the importance of this question, and the problem that this wasn’t clearer in the text.
> > We have added a short confirmation that indeed: gradient-based-trainable means there is no barren plateau.
> > In terms of precise meaning, with “a model is gradient-based trainable” we mean “a model is trainable and the learning algorithm is based on gradients”, with further details in Appendix E.
> > The proof of the statement “gradient-based trainability implies absence of BPs” is now provided by means of referencing to the textbook by Goodfellow et al. (2016), where the only required intermediate step is “if a non-trivial optimization problem can be solved via gradients, then it follows the gradient is not vanishingly small almost everywhere”.
> > We hope we have addressed the question, and we are ready to provide more details both here and in the text if needed.
> >
> > > On page 2, the risk functional is defined over the space $Y^X$.
> > > Can you explain this notation, specifically, why the data domain $X$ appears as the power in the label co-domain $Y$, but not vice versa?
> >
> > We briefly offer an explanation for this notation, but following a comment above we have removed this notation.
> > Given two sets $X$ and $Y$, the notation “$Y^X =$ {$f:X \to Y$}” is standard in some corners of applied algebra that we are accustomed to.
> > The notation arises from a parallelism between function spaces and vector spaces.
> > In a way, a vector can be seen as a function on a finite domain, where each entry is the value of the function on an element of the domain.
> > After searching online, the best alternative notations we could find are $\operatorname{Hom}(X,Y)$ or simply {$f:X \to Y$}, which we don’t feel are more advantageous than the standard $Y^X$.
> > Given that even the standard can result in confusion, we decide to avoid using any notation, and in the updated first part of Section 2 we refer to “functions from $X$ to $Y$” simply in words.
> >
> > > I am a bit uncertain about the name "vari veryational".
> > > Is there further justification for this name, besides it's a spoonerism of "very variational"?
> > > It feels a bit uninformative to the general quantum information audience.
> >
> > This is indeed an important point, we are aware our choice of name can be polarizing.
> > We aimed to find a name that captures exactly one idea: “these models are almost very variational, but not quite”, as we explain in the fourth paragraph of Section 4.1.
> > The choice of the spoonerism implicitly highlights that the name is a humorous attempt at catching the attention of the community, without losing sight of the goal: to converge to a good definition of very variational QML models.
> >
> > We have discussed among the authors what would be better alternatives, and we are ready to change the nomenclature, but until now we haven’t found any one that elegantly captures the essence of “almost very variational, but not quite”.
> > Some of these alternatives are: quasi-variational, semi-variational, almost-variational, para-variational, and similar.
> > But all of these fail on the account that these models are variational, just not very variational.
> >
> > We would be grateful to the reviewers if they offered their take on the best choice, in case a change should be made.
> >
> > Ian Goodfellow, Yoshua Bengio, and Aaron Courville. *Deep Learning*. MIT Press, (2016). http://www.deeplearningbook.org.

---

> > > ### Author Response · Authors · 2024-11-16
> > > **Overall assessment**
> > >
> > > We believe the changes prompted by the feedback from the reviewer have increased the quality of our submission in terms of clarity and presentation of our results.
> > > By clarifying the notation, we believe our text now reads nicer and is overall clearer.
> > > We hope that we addressed the concerns raised by the reviewer, both here in writing and also in the updated manuscript, and that these clarifications could help improve the opinion of the reviewer.

---

> > > > ### Comment · Reviewer_K1hL · 2024-11-27
> > > >
> > > > We thank the authors for their detailed response to the question, particularly regarding the relevance of this work in a practical setting. I believe the updated version has addressed the previously raised questions, and I have adjusted the score accordingly.

---

> > > > > ### Author Response · Authors · 2024-11-27
> > > > > **Thanking Reviewer K1hL**
> > > > >
> > > > > We thank the reviewer for their kind words and their positive evaluation of the improvements prompted from the review process.

---

### Author Response · Authors · 2024-11-16
**Updated submission, redline version**

Dear reviewers, dear Area Chairs,

Following the advice from the reviewers, we have improved our manuscript by implementing a number of changes.
We have uploaded a redline version of our updated manuscript, where the main changes are highlighted in blue.

These changes have been prompted by specific comments from the reviewers, which we briefly summarize for reference:
 - First part of Section 2: We have streamlined the notation and the presentation of supervised learning, to clear the notational clutter and center on the relevant quantities for our analysis further below.
 - Last paragraph of Section 2.4: We have added a paragraph with a brief discussion of recent advances in random-circuit-sampling and unsupervised learning.
 - Section 3.3: This is a new section that contains only the critical parts of what used to be Sections 3.3 and 3.4. This section now references Appendix C for the extended discussion, containing what used to be Sections 3.3 and 3.4, as well as new discussions on how existing QML models fit in our formalism.
 - First paragraph of Section 4: Since this is a transition paragraph that quotes Section 3, we had to make small adjustments for consistency.
 - Third paragraph of Section 4.1: We have added a proper explanation about the relation between gradient-based trainability and absence of Barren Plateaus, which was missing in the initial submission.
 - Section 5: This is a new section, containing parts of what used to be Section 5, as well as new paragraphs discussing the nature and impact of our contributions.
 - Section 6: This is a new section, containing parts of what used to be Section 6, but especially serving the purpose of separating the summary from the discussion. We have added a paragraph with closing remarks about the broad applicability of our results.
 - Appendix C: This is a new appendix, containing what used to be Sections 3.3 and 3.4, as well as a new Section C.3 discussing how other existing QML models fit our formalization.

We hope this overview of changes aids in highlighting our earnest effort in improving our manuscript following the advice and comments by the reviewers, which we found invariably helpful and valuable.

---

### Meta-Review · Area_Chair_PPA5 · 2024-12-19

**Metareview:**

This paper investigates the compatibility of trainability and non-dequantizability in variational quantum machine learning (QML) models. The authors introduce the concept of "vari veryational" models, providing a theoretical framework and constructions that meet these two criteria. While reviewers raised concerns about the limited practical implications of the constructions, the focus on supervised learning, and the somewhat contrived nature of the proposed models, the work offers valuable insights into the design of variational QML models and advances understanding in the field.

**Additional Comments On Reviewer Discussion:**

The authors and reviewers engaged in discussions addressing several critical points. These included the limited practical implications of the proposed "vari veryational" QML models, the specificity of the results to supervised learning, and the somewhat contrived nature of the constructions. Reviewers also raised concerns about the clarity of the paper's broader impact on the design of variational QML models and the potential limitations of focusing primarily on gradient-based training and classical hardness.

During the rebuttal period, the authors provided detailed responses, clarifying their theoretical framework and emphasizing the importance of resolving the compatibility of trainability and non-dequantizability as foundational questions in QML. They streamlined sections of the paper to improve clarity, expanded the discussion of related work, and addressed the potential for generalizing their constructions to other settings.

---

### Decision · Program_Chairs · 2025-01-22

Accept (Poster)